# RPL3L-containing ribosomes determine translation elongation dynamics required for cardiac function

Chisa Shiraishi[1,7], Akinobu Matsumoto[1,7] ✉, Kazuya Ichihara[1], Taishi Yamamoto[2], Takeshi Yokoyama[3], Taisuke Mizoo[1], Atsushi Hatano[4], Masaki Matsumoto[4], Yoshikazu Tanaka[3], Eriko Matsuura-Suzuki[5], Shintaro Iwasaki[5,6], Shouji Matsushima[2], Hiroyuki Tsutsui[2] & Keiichi I. Nakayama[1] ✉

Although several ribosomal protein paralogs are expressed in a tissue-specific manner, how these proteins affect translation and why they are required only in certain tissues have remained unclear. Here we show that RPL3L, a paralog of RPL3 specifically expressed in heart and skeletal muscle, influences translation elongation dynamics. Deficiency of RPL3L-containing ribosomes in RPL3L knockout male mice resulted in impaired cardiac contractility. Ribosome occupancy at mRNA codons was found to be altered in the RPL3L-deficient heart, and the changes were negatively correlated with those observed in myoblasts overexpressing RPL3L. RPL3L-containing ribosomes were less prone to collisions compared with RPL3-containing canonical ribosomes. Although the loss of RPL3L-containing ribosomes altered translation elongation dynamics for the entire transcriptome, its effects were most pronounced for transcripts related to cardiac muscle contraction and dilated cardiomyopathy, with the abundance of the encoded proteins being correspondingly decreased. Our results provide further insight into the mechanisms and physiological relevance of tissue-specific translational regulation.

Information encoded by an mRNA is translated by ribosomes for production of the corresponding protein. The ribosome is a ribonucleoprotein complex that in eukaryotes consists of a small 40S subunit composed of 33 ribosomal proteins (RPs) and 18S rRNA as well as a large 60S subunit composed of 47 RPs and 5S, 5.8S, and 28S rRNAs[1]. Although ribosomes have generally been considered to be homogeneous with regard to their structure and organization, recent studies have revealed the existence of differences in ribosome organization and posttranslational modifications and have led to the proposal of concepts such as "ribosome heterogeneity" and a "ribosome code" analogous to the histone code hypothesis[2–6]. Such concepts are based on the notion that this heterogeneity gives rise to specific translational regulation that adds additional complexity to the control of gene expression and which is important for a variety of physiological processes. For example, ribosomes in mouse embryonic stem cells show a heterogeneous RP composition, with those containing RPS25/eS25 or RPL10A/uL1 preferentially translating specific subsets of mRNAs that contribute to metabolism, the cell cycle, and development[7].

[1]Division of Cell Biology, Medical Institute of Bioregulation, Kyushu University, Fukuoka, Fukuoka 812-8582, Japan. [2]Department of Cardiovascular Medicine, Faculty of Medical Sciences, Kyushu University, Fukuoka, Fukuoka 812-8582, Japan. [3]Graduate School of Life Sciences, Tohoku University, Sendai, Miyagi 980-8577, Japan. [4]Department of Omics and Systems Biology, Graduate School of Medical and Dental Sciences, Niigata University, Niigata, Niigata 951-8510, Japan. [5]RNA Systems Biochemistry Laboratory, RIKEN Cluster for Pioneering Research, Wako, Saitama 351-0198, Japan. [6]Department of Computational Biology and Medical Sciences, Graduate School of Frontier Sciences, The University of Tokyo, Kashiwa, Chiba 277-8561, Japan. [7]These authors contributed equally: Chisa Shiraishi, Akinobu Matsumoto. ✉e-mail: akinobu@bioreg.kyushu-u.ac.jp; nakayak1@bioreg.kyushu-u.ac.jp

Phosphorylation of the Ser[38] residue of RPL12/uL11 is also more prevalent in monosomes than in polysomes and affects cell cycle progression by regulating the translation of a specific subset of mitosis-related mRNAs[8].

Many RP paralogs have been identified in eukaryotes, some of which are expressed in a tissue-specific manner[9]. In mammals, RPL10L/uL16L and RPL39L/eL39L show testis-specific expression, and expression of RPL3L is restricted to heart and skeletal muscle[10–14]. Male meiotic cells inactivate both X and Y chromosomes, a process termed meiotic sex chromosome inactivation[15], and expression of RPL10L and RPL39L (which are encoded by somatic chromosomes) may compensate for the deficiency of RPL10 and RPL39 (which are encoded by the X chromosome) during this process[16,17]. In contrast to the situation for the testis-specific RP paralogs, expression of RPL3/uL3 is maintained in skeletal and cardiac muscle, albeit at a lower level than in other tissues, suggesting that expression of RPL3L might in part serve to complement this low level of RPL3 expression in these tissues[14]. Although their amino acid sequences are similar overall and the proteins likely occupy the same position in the ribosome, RPL3L and RPL3 may have different functions to some extent given that they differ at about one-quarter of their amino acid positions[18]. RPL3L expression is downregulated during induction of skeletal muscle hypertrophy in mice[19]. Forced expression of RPL3L in C2C12 myogenic cells inhibits myoblast fusion and myotube growth[19], whereas loss of RPL3L increases ribosome-mitochondrion interactions in mouse cardiomyocytes[20]. Furthermore, mutations of the human *RPL3L* gene have been detected in individuals with atrial fibrillation or neonatal dilated cardiomyopathy[21–25].

RPL3 is evolutionarily conserved and is the RP located closest to the peptidyl transferase center (PTC) of the ribosome[26]. It folds as a globular domain and possesses three fingerlike structures that are positioned near the A-site tRNA (A-tRNA) binding pocket of the PTC and are known as the NH₂-terminal extension, the Trp finger (W finger), and the basic thumb, with both the W finger and the basic thumb being part of a central loop[26]. A W255C mutation in the W finger of yeast Rpl3 affects the conformation of the A-tRNA binding pocket of the PTC, resulting in reduced peptidyl transferase activity[26,27]. In addition, His[245] of human RPL3 is methylated by the methyltransferase METTL18, and defective methylation results in an increased rate of elongation at Tyr codons and aggregation of Tyr-rich proteins[28,29].

In the present study, we generated RPL3L-deficient mice and found that loss of RPL3L results in impaired cardiac contractility. Translation elongation dynamics were shown to be altered and the frequency of ribosome collisions to be increased in the RPL3L-null heart, resulting in reduced expression of proteins that contribute to cardiac function. Our data indicate that RPL3L-containing ribosomes (RPL3L-ribosomes) are required for proper cardiac function as a result of their regulation of translation elongation dynamics.

## Results

### Expression pattern and predicted 3D structure of RPL3L
RPL3L was previously shown to be expressed specifically in heart and skeletal muscle[18], and we validated this finding by analysis of RNA-sequencing (RNA-seq) data for a wider range of human tissues from the Genotype-Tissue Expression (GTEx) database. We thus found that *RPL3L* was indeed highly expressed only in heart and skeletal muscle, whereas *RPL3* was ubiquitously expressed, although its expression level in heart and skeletal muscle was lower than that in other tissues (Fig. 1a). RNA-seq data for mouse tissues confirmed a similar expression pattern for *Rpl3l* and *Rpl3* (Supplementary Fig. 1a)[30,31]. Further analysis of publicly available single-cell RNA-seq (scRNA-seq) data for the human heart revealed that *RPL3L* expression was largely restricted to cardiomyocytes (Fig. 1b, c), with many cardiomyocytes expressing both *RPL3L* and *RPL3* (Fig. 1d, e)[32]. Similar results were obtained with publicly available scRNA-seq data for the mouse heart (Supplementary Fig. 1b–e)[33].

The binding mode of RPL3 was investigated with the three-dimensional (3D) structural model of the human 80S ribosome determined by single-particle cryo-electron microscopy (cryo-EM) (PDB ID: 6IP5)[34]. The COOH-terminal region of RPL3 engages in several interactions with RNA helices exposed to the solvent, whereas the NH₂-terminal region of the protein is extended toward the core region of the 60S subunit (Fig. 1f, g). The NH₂-terminal tip of RPL3 is exposed to the intersubunit cavity and is situated in the vicinity of helix 92 of 28S rRNA, which may affect the amino acid moieties of aminoacyl-tRNAs during their accommodation to the A-site. To compare the structures of RPL3L and RPL3, we obtained a predicted model of RPL3L from the AlphaFold protein structure database and superimposed it on RPL3 in the 80S ribosome (Fig. 1g)[35,36]. The COOH-terminus of RPL3L is extended by four amino acid residues compared with RPL3, as shown previously[20]. Although the structural organization of the three fingerlike projections of RPL3 and of RPL3L was similar, several amino acids of RPL3L in these regions differ from those of RPL3 and these amino acids of RPL3L are well conserved among vertebrates (Fig. 1h), suggesting that these local differences between RPL3 and RPL3L may affect the A-tRNA binding pocket of the PTC. However, this issue warrants verification by cryo-EM analysis of RPL3L-ribosomes, given that the structure of RPL3L is only a prediction by AlphaFold.

### Impaired cardiac contractility in mice lacking RPL3L-ribosomes
To clarify the physiological role of RPL3L, we generated mice deficient in RPL3 or RPL3L with the use of the CRISPR-Cas9 system, with deletion of exon 2 giving rise to a frameshift mutation in both types of mutant mice (Supplementary Fig. 2a, b). Mating of corresponding heterozygous mutant mice yielded weaned mice with homozygous mutation of *Rpl3l* (*Rpl3l*⁻/⁻) in approximately the expected Mendelian ratio but no *Rpl3* homozygous mutant (*Rpl3*⁻/⁻) mice, suggesting that loss of RPL3 results in embryonic death (Fig. 2a). Reverse transcription (RT) and quantitative polymerase chain reaction (qPCR) analysis of the heart and skeletal muscle confirmed that *Rpl3l* mRNA was either undetectable for the deleted region or reduced in abundance for the 3′ untranslated region (3′UTR), likely as a result of nonsense-mediated mRNA decay (Fig. 2b). Of note, ablation of RPL3L resulted in upregulation of *Rpl3* mRNA, suggestive of a compensatory response to the loss of RPL3L function, with similar compensatory expression having previously been described for RPL22 and RPL22L1[37]. We further confirmed the absence of RPL3L protein in the heart of *Rpl3l*⁻/⁻ mice by mass spectrometry (MS)–based multiple reaction monitoring (MRM) (Fig. 2c)[38]. Given that MRM analysis can measure absolute amounts of proteins, we determined the amounts of individual RPs and normalized them by the average of the amounts of all RPs in order to calculate the number of RP molecules per 80S ribosome. This analysis revealed that RPL3L-ribosomes and RPL3-containing canonical ribosomes were present at a ratio of ~2:1 in the heart of *Rpl3l*⁺/⁺ mice. Loss of RPL3L resulted in an increased protein level for RPL3 in ribosomes, with no significant difference in that for other RPs (Supplementary Fig. 2c). Although RPS15 abundance tended to be increased in the RPL3L-deficient heart relative to the control heart, this difference was not statistically significant. It is also unlikely that the loss of RPL3L would affect the amount of RPS15, given the distance between the two proteins in the 3D structure of the ribosome.

*Rpl3l*⁻/⁻ mice manifested no significant difference in body weight or in the weight of the heart or skeletal muscle—including the tibialis anterior (TA), extensor digitorum longus (EDL), gastrocnemius (GC), and soleus (Sol)—compared with *Rpl3l*⁺/⁺ mice (Fig. 2d, e). Histopathologic analysis revealed no obvious changes in myofiber size as determined by wheat germ agglutinin (WGA) staining or in the extent of cardiac fibrosis as visualized by Masson's trichrome staining in the heart of *Rpl3l*⁻/⁻ mice compared with that of control mice (Fig. 2f, g). However, echocardiographic profiles revealed that left ventricular

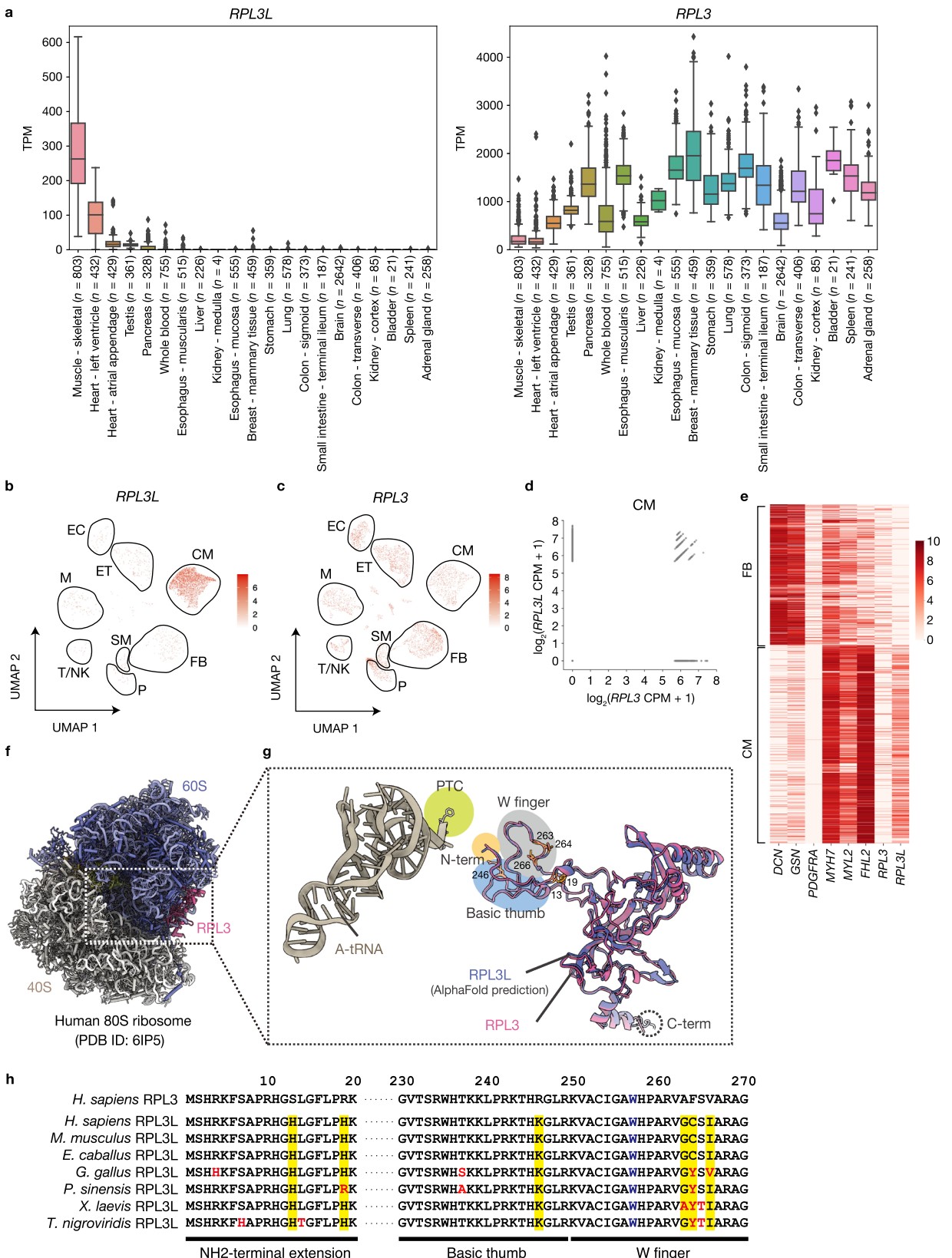

fractional shortening (LVFS) and the left ventricular ejection fraction (LVEF), both of which are key indicators of cardiac contractility, were significantly reduced by the loss of RPL3L, whereas the left ventricular end-systolic diameter (LVDs) was increased and left ventricular wall thickness (LVWT) and the left ventricular end-diastolic diameter (LVDd) were unchanged (Fig. 2h, i, Supplementary Data 1, and

Supplementary Movie 1). These results suggested that loss of RPL3L-ribosomes results in reduced cardiac contractility.

We also examined the effects of cardiac pressure overload-induced by angiotensin (Ang) II infusion with a micro−osmotic pump. Cardiac hypertrophy, especially of the left ventricle, was apparent after Ang II infusion, confirming the induction of cardiac pressure

**Fig. 1 | Expression pattern and predicted structure of RPL3L. a** Expression patterns of *RPL3L* and *RPL3* in human tissues. The RNA-seq data were obtained from the GTEx portal and are presented as box-and-whisker plots, with the boxes representing the median and upper and lower quartiles and the whiskers representing the maximum and minimum values. Diamond-shaped dots indicate outliers. TPM, transcripts per million. **b, c** UMAP (Uniform Manifold Approximation and Projection) plots from Seurat analysis of scRNA-seq data for *RPL3L* and *RPL3* expression in the human heart. Data are from GSE183852. Groups corresponding to cardiomyocytes (CM), smooth muscle (SM), fibroblasts (FB), pericytes (P), T cells and natural killer cells (T/NK), macrophages and monocytes (M), endothelium (ET), and endocardium (EC) are enclosed in black frames. **d** Expression levels of *RPL3* and *RPL3L* in single cells for cardiomyocytes as in **b** and **c**. CPM, counts per million. **e** Heat map showing row-scaled expression of the marker genes for FB and CM as well as of *RPL3* and *RPL3L* for the FB and CM clusters in **b** and **c**. **f, g** Structural comparison for RPL3 and RPL3L. The atomic model of RPL3 is highlighted (magenta) in the model of the human 80S ribosome obtained by cryo-EM (PDB ID: 6IP5) (**f**). The three fingerlike projections—the $NH_2$-terminal extension (N-term), the W finger, and the basic thumb—of RPL3/RPL3L extend into the core region of the 60S subunit and are situated in close proximity to the CCA terminus of the A-tRNA in the PTC (**g**). The amino acids that differ between RPL3 and RPL3L is highlighted in orange with their amino acid numbers. **h** Comparison of amino acid sequences of human (*Homo sapiens*) RPL3 with those of human RPL3L as well as of RPL3L from other mammalian (*Mus musculus* and *Equus caballus*), avian (*Gallus gallus*), reptile (*Pelodiscus sinensis*), amphibian (*Xenopus laevis*), and fish (*Tetraodon nigroviridis*) species. Amino acids that differ between human RPL3 and the various RPL3L proteins are highlighted in yellow, and those of RPL3L that differ among species are shown in red. The W residue in blue corresponds to W255 of yeast Rpl3.

overload, but there was no significant difference in the change in heart or LV weight observed in response to Ang II treatment between RPL3L-deficient and control mice (Supplementary Fig. 3a). In addition, whereas LVEF and LVFS were significantly reduced in RPL3L-deficient mice compared with *Rpl3l*$^{+/+}$ mice treated with saline, these differences were no longer significant for the mice treated with Ang II (Supplementary Fig. 3b, c, and Supplementary Data 1). These results indicated that the reduced cardiac contractility of mice lacking RPL3L was not exacerbated by pressure overload-induced by Ang II stimulation.

### Altered translation elongation dynamics in the RPL3L-deficient heart

To explore the specific role of RPL3L-ribosomes in translation, we performed Ribo-seq analysis with heart lysates from *Rpl3l*$^{+/+}$ and *Rpl3l*$^{-/-}$ mice[39]. The results revealed high 3−nucleotide (nt) periodicity scores for the Ribo-seq libraries prepared from each genotype (Supplementary Fig. 4a, b)[40]. No transcripts manifested a significant change in translation efficiency (TE: Ribo-seq reads normalized by RNA-seq reads) in response to the loss of RPL3L (Fig. 3a and Supplementary Data 2). Although the number of ribosome footprints derived from *Rpl3l* transcripts was reduced, the corresponding RNA abundance was also reduced by almost the same extent, with TE thus remaining unchanged (Fig. 3b).

Despite the lack of significant changes in TE, the RPL3L-ribosome−deficient heart showed delayed translational elongation at codons for Pro and Ala and a slightly increased rate of elongation at codons for Asn and Lys, as indicated by increased ribosomal occupancy for Pro and Ala codons at the A-site and by reduced ribosomal occupancy for Asn and Lys codons at this site, respectively (Fig. 3c, d). Ribosomal occupancy showed similar trends, but to a lesser extent, at the P- and E-sites (Supplementary Fig. 4c, d). At the A-site, ribosome occupancy was increased for all Pro and Ala codons and decreased for all Lys and Asn codons (Fig. 3e). Further analysis with the use of RUST, which reduces the impact of data heterogeneity by a simple normalization method[41], confirmed an increase in ribosome occupancy at all Pro and Ala codons, but failed to show a decrease in ribosome occupancy at Lys and Asn codons (Fig. 3f), suggesting that the delayed translational elongation at Pro and Ala codons is a more robust abnormality in the RPL3L-deficient heart.

The rate of translation elongation is not constant even under normal translation conditions, with ribosomes undergoing local acceleration or deceleration to ensure proper translation. An abnormal delay in elongation results in a reduced level of protein synthesis and in ribosome collision, whereas an aberrantly increased rate of elongation prevents proper folding of the nascent protein and results in the formation of protein aggregates[29,42,43]. We next analyzed the coefficient of variation (CV) for ribosome occupancy to determine whether RPL3L deficiency affects overall translation elongation dynamics. An increased or decreased rate of translation elongation at certain positions would increase the CV of ribosome occupancy at the A-site (Fig. 3g). In contrast, the CV of ribosome occupancy was shown to be decreased in the cerebrum of FMR1-null mice as a result of a widespread reduction in translational pauses[44]. We observed a significant increase in the CV of ribosome occupancy at the A-site in association with the loss of RPL3L in the heart (Fig. 3h), indicating that RPL3L-ribosomes affect translation elongation dynamics.

### Negative correlation of changes in ribosome occupancy induced by loss or overexpression of RPL3L

To further validate the contribution of RPL3L to translation elongation dynamics, we infected mouse C2C12 myoblasts with a retrovirus encoding mouse RPL3 or RPL3L (RPL3 OE and RPL3L OE cells, respectively). *Rpl3l* mRNA was virtually undetectable in C2C12 cells, and overexpression of RPL3L resulted in downregulation of *Rpl3* mRNA (Fig. 4a), an effect opposite to the upregulation apparent in the RPL3L-deficient heart (Fig. 2b). With the use of a primer set targeting the 3′UTR of *Rpl3* mRNA, we found that the abundance of endogenous *Rpl3* mRNA was also reduced by overexpression of RPL3 (Fig. 4a). In addition, a luciferase reporter assay revealed that *Rpl3* promoter activity was suppressed to similar extents in both RPL3 OE and RPL3L OE cells (Supplementary Fig. 5a). These results indicated that RPL3L suppresses the expression level of *Rpl3* mRNA, but that this activity is identical for RPL3L and RPL3.

MRM analysis revealed that RPL3L-ribosomes and RPL3-containing canonical ribosomes were present at a ratio of ~3:2 in RPL3L OE cells (Fig. 4b). Ribo-seq analysis of these infected cells revealed marked changes in TE for many transcripts in RPL3L OE cells, whereas the changes were much less pronounced in RPL3 OE cells (Supplementary Fig. 5b and Supplementary Data 3). We also found that overexpression of RPL3L resulted in reduced ribosome occupancy for Pro and Ala codons and increased ribosome occupancy for Lys codons at the A-site (Fig. 4c, d, and Supplementary Fig. 5c), which are opposite changes to those apparent for the RPL3L-deficient heart (Fig. 3c, d). Again, the changes in ribosome occupancy induced by overexpression of RPL3 were much smaller than those resulting from overexpression of RPL3L (Fig. 4c, d, and Supplementary Fig. 5d). The changes in ribosome occupancy at the E- and P-sites were also small for both RPL3 OE and RPL3L OE cells (Supplementary Fig. 5e, f). Of note, the changes in ribosome occupancy at the A-site in RPL3L OE cells showed a marked negative correlation with those in the RPL3L-deficient heart, with no or a less pronounced negative correlation being apparent for the P- and E-sites, respectively (Fig. 4e). Furthermore, no correlation was detected for the changes in ribosome occupancy between RPL3 OE cells and the RPL3L-deficient heart at A-, P-, or E-sites (Fig. 4f). Although overexpression of RPL3 reduced the CV of ribosome occupancy at the A-site relative to that for control cells, overexpression of RPL3L led to a greater such reduction (Fig. 4g). These results thus further confirmed that RPL3L-ribosomes are required for regulation of translation elongation dynamics in the heart.

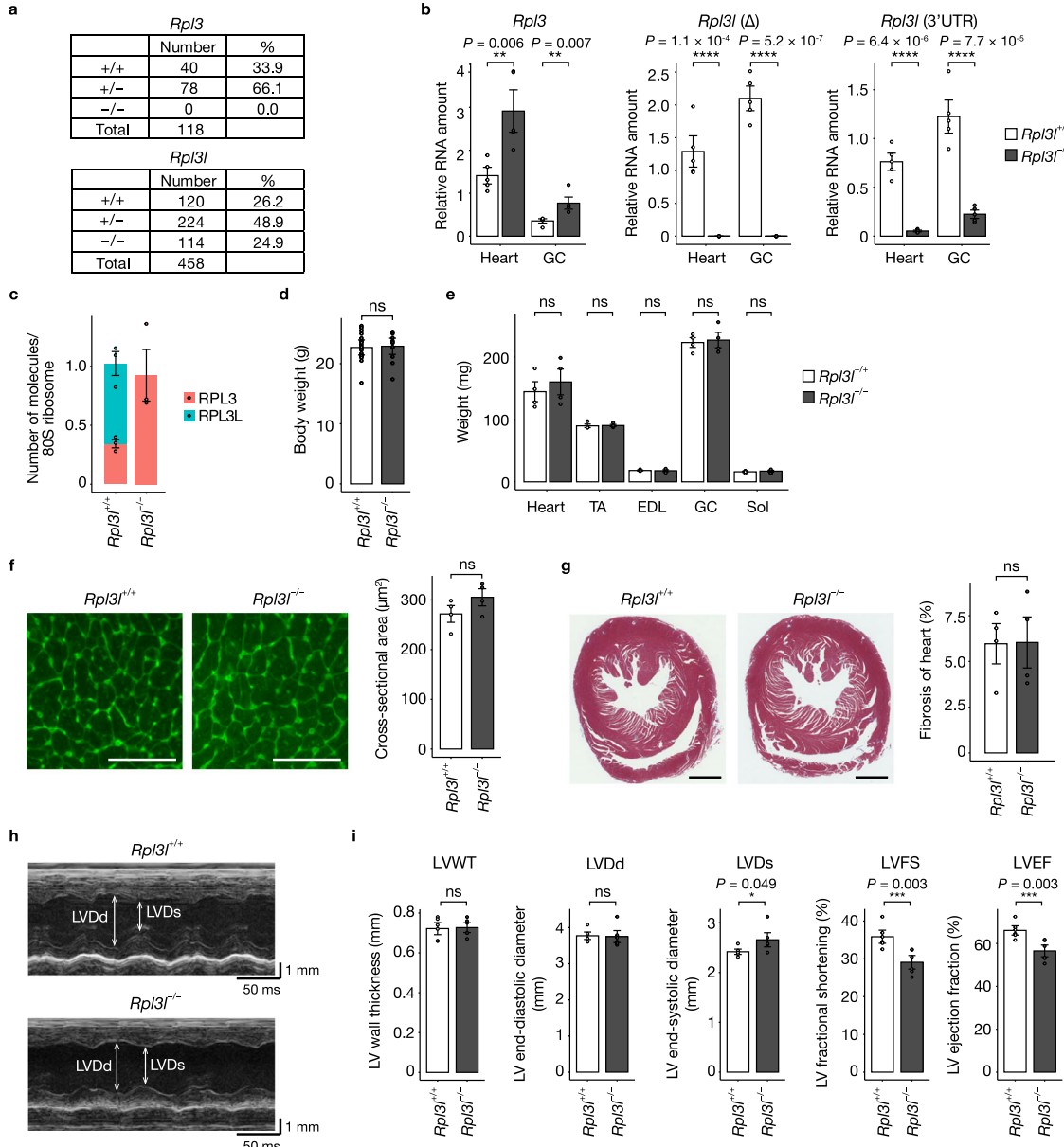

**Fig. 2 | RPL3L-deficient mice manifest impaired cardiac contractility. a** Number and percentage of mice of the indicated genotypes at weaning for the offspring produced by mating male and female heterozygotes for each mutation. **b** RT-qPCR analysis of *Rpl3l* and *Rpl3* mRNA abundance in the heart and gastrocnemius (GC) of *Rpl3l*[+/+] and *Rpl3l*[−/−] mice at 10 weeks of age (*n* = 5 mice). *Rpl3l* transcripts were measured with primers targeting either the deleted region (Δ) or the intact 3′UTR. **c** Number of RPL3 and RPL3L molecules per 80S ribosome as determined by MRM analysis of the polysome fraction from the heart of *Rpl3l*[+/+] and *Rpl3l*[−/−] mice at 10 weeks of age (*n* = 3 mice). **d, e** Body weight of *Rpl3l*[+/+] (*n* = 21 mice) and *Rpl3l*[−/−] (*n* = 12 mice) mice at 10 weeks of age and tissue weight for the heart as well as TA, EDL, GC, and Sol muscles of *Rpl3l*[+/+] and *Rpl3l*[−/−] mice (*n* = 4 mice) at 18 to 19 weeks of age (**e**). **f, g** Representative images of WGA staining and Masson's trichrome staining for heart sections from *Rpl3l*[+/+] and *Rpl3l*[−/−] mice at 18 to 19 weeks of age (left panels) and quantitative analysis of muscle fiber size and the extent of fibrosis (blue staining) determined from such sections, respectively (*n* = 4 mice) (right panels). Scale bars, 50 μm (**f**) and 1 mm (**g**). **h, i** Representative echocardiographic images and quantitative analysis of LVWT, LVDd, LVDs, LVFS, and LVEF for *Rpl3l*[+/+] and *Rpl3l*[−/−] mice at 18 to 19 weeks of age (*n* = 6 mice). All quantitative data in bar graphs are means ± s.d. *P < 0.05, **P < 0.01, ***P < 0.005, ****P < 0.001; ns not significant (unpaired two-tailed Student's *t* test). Source data are provided as a Source Data file.

Whereas forced expression of RPL3L in C2C12 myogenic cells was previously shown to inhibit myoblast fusion and myotube growth[19], our Ribo-seq analysis of C2C12 myoblasts was performed under the proliferative condition rather than the differentiation condition. To examine further the effects of RPL3 or RPL3L overexpression under the proliferative condition, we performed gene ontology (GO) analysis of genes upregulated in RPL3L OE cells compared with RPL3 OE cells with the use of RNA-seq data. We found that genes for "regulation of muscle contraction" and "skeletal muscle contraction" were upregulated in RPL3L OE cells relative to RPL3 OE cells (Supplementary Fig. 6a), consistent with the echocardiography results obtained for the RPL3L-deficient heart (Fig. 2i). Of note, genes for "skeletal muscle cell differentiation" were also upregulated in RPL3L OE cells compared with RPL3 OE cells (Supplementary Fig. 6a), which appears inconsistent with the previous finding that overexpression of RPL3L in C2C12 myoblasts inhibited myoblast fusion and myotube growth[19]. Indeed, we observed significant attenuation of myoblast fusion and myotube growth in RPL3 OE cells compared with control cells, but not in RPL3L

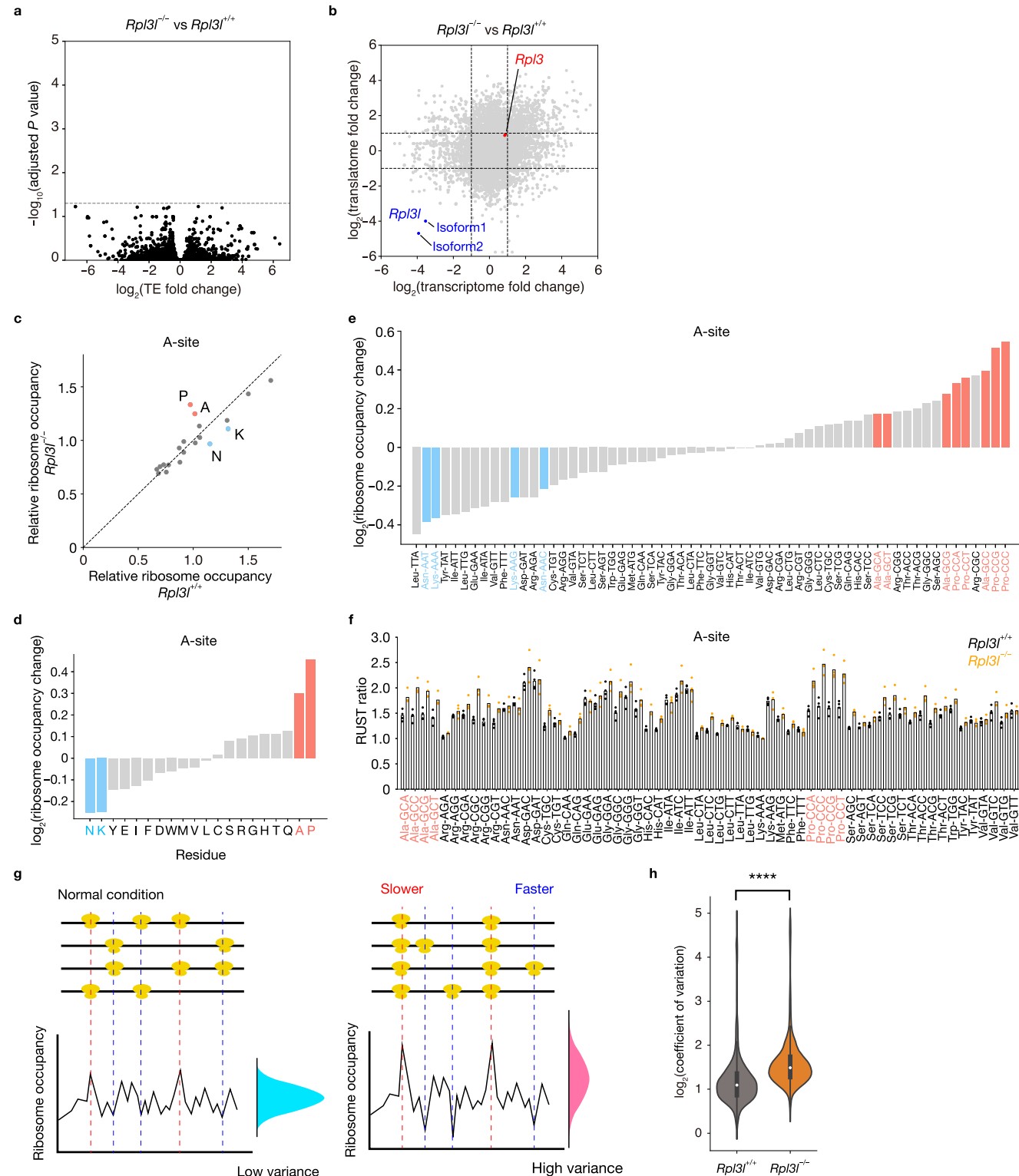

OE cells (Supplementary Fig. 6b, c). These results indicated that overexpression of RPL3, but not that of RPL3L, suppresses myoblast differentiation. Overall, our findings with C2C12 myoblasts further corroborated those with RPL3L-deficient mice.

### Reduced protein expression associated with aberrant translation elongation dynamics

Although elongation dynamics for the overall transcriptome were affected by the loss of RPL3L, the extent of the effect might vary among transcripts. We therefore analyzed further the fold change in

ribosome occupancy at the A-site in the RPL3L-deficient heart. Ribosome occupancy for the 41 Pro codons in 28 genes and the 51 Ala codons in 20 genes was significantly increased compared with the control heart (Fig. 5a, Supplementary Fig. 7, and Supplementary Data 4). Metagene plot analysis of the Ribo-seq data confirmed the accumulation of footprint reads around Pro and Ala codons for genes with significantly increased ribosome occupancy at the A-site for such codons (Fig. 5b).

To clarify the function of such genes, we performed GO terminology analysis, which revealed enrichment of terms related to cardiac

**Fig. 3 | Altered elongation dynamics in the RPL3L-ribosome–deficient heart.**
**a** Volcano plot for differential TE as determined by Ribo-seq and RNA-seq analysis
($n = 4$ mice). The change in TE for each transcript in the heart of $Rpl3l^{-/-}$ mice at
10 weeks of age relative to that for $Rpl3l^{+/+}$ mice was analyzed with RiboDiff. $P$ values
were calculated by chi-square test and adjusted by Benjamini-Hochberg method for
multiple comparisons. The gray dashed line indicates an adjusted $P$ value of 0.05.
**b** Fold change in the number of RNA-seq reads (transcriptome) and Ribo-seq
footprint reads (translatome) for the $Rpl3l^{-/-}$ heart compared with the control heart
($n = 4$ mice). The gray dashed lines indicate a $\log_2$(fold change) of ±1. $Rpl3l$ isoform1:
ENSMUST00000045186.10. $Rpl3l$ isoform2: ENSMUST00000170239.8. **c** Relative
ribosome occupancy at A-site codons in the heart of $Rpl3l^{+/+}$ and $Rpl3l^{-/-}$ mice ($n = 4$
mice). Data were aggregated according to all codons for each amino acid.
**d**–**f** Ribosome occupancy changes at A-site codons for each amino acid residue or

codon, as well as RUST ratio values at A-site codons, in the heart of $Rpl3l^{-/-}$ mice
compared with that of control mice ($n = 4$ mice). **g** Schematic representation of the
CV for ribosome occupancy at the A-site. A delay in translation elongation at a
particular codon results in an increase in ribosome occupancy at the A-site, whereas
an increase in elongation rate results in a decrease in ribosome occupancy at the
A-site. Changes in elongation dynamics therefore lead to changes in the magnitude
of the variance of ribosome occupancy, as indicated by the blue and red curves.
**h** Violin plots for the CV of ribosome occupancy at the A-site in the heart of $Rpl3l^{+/+}$
and $Rpl3l^{-/-}$ mice ($n = 4$ mice). The inner boxes represent the median and upper and
lower quartiles, and the whiskers represent the maximum and minimum values.
***$P < 0.001$ (two-tailed Mann-Whitney $U$ test). Source data are provided as a
Source Data file.

muscle contraction and mitochondrial ATP synthesis (Fig. 5c). For
example, both *Myl4* (which encodes a myosin light chain) and *Atp5o*
(which encodes a subunit of the mitochondrial $F_1$ complex for ATP
synthesis) showed altered translation elongation dynamics in response
to the loss of RPL3L (Fig. 5d). We therefore performed quantitative
proteomics analysis by MS with data-independent acquisition (DIA) to
investigate whether the abnormal elongation dynamics for the iden-
tified genes affected the abundance of the corresponding proteins.
The RPL3L-deficient heart showed reduced protein levels for genes
with delayed translation elongation at Pro/Ala codons (Fig. 5e and
Supplementary Data 5). These results suggested that certain genes are
susceptible to regulation of translational dynamics by RPL3L-
ribosomes and that the encoded proteins are more efficiently pro-
duced by such ribosomes.

## Abundance and charge status of tRNAs and m6A status of mRNAs in the RPL3L-deficient heart

Given that tRNA status is also a key determinant of translation elon-
gation, we next analyzed the abundance and aminoacylation status of
tRNAs by Charged DM-tRNA-seq[45]. We detected a decrease in the
amount of tRNA-Pro-CGG and an increase in that of tRNA-Pro-TGG,
whereas the amounts of other tRNAs remained unchanged, in the
RPL3L-deficient heart compared with the control heart (Fig. 6a, b, and
Supplementary Data 6). The rate of aminoacylation of all tRNAs, as well
as the protein levels of aminoacyl-tRNA synthetases, were not altered
in the RPL3L-deficient heart (Fig. 6c, d, and Supplementary Data 6).
These results suggested that the delay in translation elongation at the
CCG codon induced by the loss of RPL3L was attributable in part to a
decrease in the abundance of tRNA-Pro-CGG, whereas the delay at
other Pro and Ala codons was independent of the amount or ami-
noacylation status of tRNAs.

Given that N6-methyladenosine (m6A) modification in the coding
region alters the secondary structure of certain mRNAs and reduces
ribosomal pausing[46], we also performed m6A-seq for the RPL3L-
deficient and control heart. No significant change in the ratio of m6A
for all transcripts (Fig. 6e, f, and Supplementary Data 7) or in the
abundance of proteins encoded by m6A-positive mRNAs (Fig. 6g) was
apparent for the RPL3L-deficient heart compared with the control
heart, suggesting that m6A status is not likely a major determinant of
the altered translation dynamics induced by the loss of RPL3L in
the heart.

## RPL3L-ribosomes are less prone to collisions

Prolonged translational pausing gives rise to ribosome collision and
the accumulation of truncated nascent proteins, which are toxic and
therefore degraded by the ribosome-associated quality control (RQC)
pathway[47,48]. The collision of stalled ribosomes with subsequent ribo-
somes results in the formation of diribosomes (disomes), which are
recognized by the RQC pathway, resulting in their dissociation and
degradation of the truncated nascent proteins. Given that loss of
RPL3L was found to affect translation elongation dynamics, we next

evaluated collisions of RPL3L-ribosomes by disome profiling (Disome-
seq) of the RPL3L-deficient and control mouse heart[49,50]. A total of 722
disome peaks was identified in the RPL3L-deficient and control heart,
with most of these peaks (652 peaks) being commonly detected and 25
and 45 peaks detected specifically in the control and RPL3L-deficient
heart, respectively (Fig. 7a). Of the 92 Pro/Ala codons that showed
delayed translation elongation in the RPL3L-deficient heart, 39 codons
were associated with disome peaks (Fig. 7b).

Given the large number of disome peaks in addition to those
associated with the 39 Pro/Ala codons (Fig. 7b), we next examined the
contribution of RPL3L-ribosomes to overall collisions. In the region of
disome peaks, an increase in disome footprint reads was observed in
association with the loss of RPL3L-ribosomes (Fig. 7c and Supple-
mentary Data 8). GO term analysis revealed that the genes with the
disome peaks also showed enrichment for terms related to mito-
chondrial ATP synthesis and cardiac muscle contraction (Fig. 7d). In
addition, protein expression for such genes was reduced in the RPL3L-
deficient heart compared with the control heart (Fig. 7e). These results
indicated that RPL3L-ribosomes influence translation elongation
dynamics not only at Pro/Ala codons but also at collision-prone sites
in the heart.

## Decreased expression of proteins related to cardiac muscle contraction in the RPL3L-deficient heart

Given the defects in cardiac contractility in RPL3L-defiient mice (Fig. 2),
we focused genes related to cardiac muscle contraction. The group of
genes with the GO term "cardiac muscle contraction" ($n = 46$) was
further explored together with those for the categories of "dilated
cardiomyopathy (DCM)–related genes"[51] ($n = 33$), "genes with differ-
ential ribosome occupancy at Pro/Ala codons in the heart of $Rpl3l^{+/+}$ or
$Rpl3l^{-/-}$ mice (Pro/Ala)" ($n = 44$), and "genes with disome peaks in the
heart of $Rpl3l^{+/+}$ or $Rpl3l^{-/-}$ mice (disome-positive)" ($n = 198$). A Venn
diagram shows the overlap of these four groups of genes (Fig. 8a).
About one-third of the genes in the "cardiac muscle contraction" group
were included in the "disome-positive" group ($n = 14$), with these
overlapping genes also including genes in the "Pro/Ala" group ($n = 5$).
These 14 genes included those for cardiac α-actin (*ACTC1*) and cardiac
myosins (*MYH6*, *MYH7*, *MYL2*, *MYL3*, and *MYL4*), which align to form
sarcomeres; the tropomyosin α1 chain (*TPM1*); the three troponin
subunits (*TNNC1*, *TNNI3*, and *TNNT2*); MYBPC3, a protein that binds to
actin and myosin and enhances cardiac contractility[52]; TCAP, a protein
that cross-links Titin, a filamentous protein with springlike properties
in the I-band region, to the Z-disc[53]; CSRP3, a scaffold protein involved
in multiple protein-protein interactions within the Z-disc[54]; and CASQ2,
a calcium-binding protein that localizes to the sarcoplasmic reticulum
to store calcium[55]. In addition, the overlap between DCM-related and
disome-positive genes included *LDB3*, which encodes a PDZ-LIM
domain–binding factor that provides structural stability to the Z-disc[56].
The cumulative fraction for the fold change in protein abundance in
the $Rpl3l^{-/-}$ mouse heart compared with the control mouse heart
(Fig. 8b) revealed that the abundance of proteins encoded by these 15

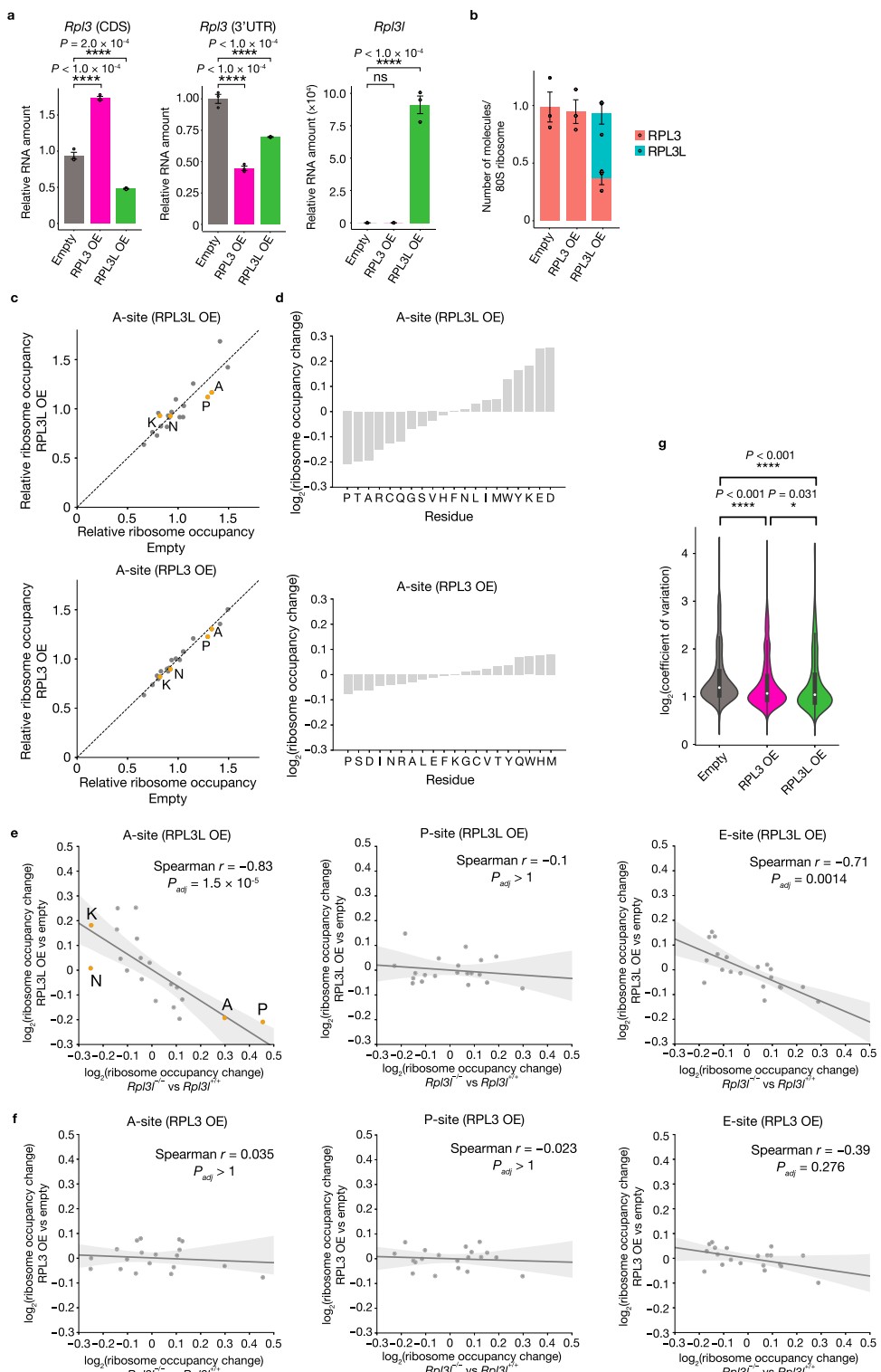

genes (enclosed by the red boundaries in Fig. 8a, although only 12 of the proteins were detected by MS) was significantly ($P = 3.03 \times 10^{-5}$) reduced in the heart of $Rpl3l^{-/-}$ mice relative to that of all proteins ($n = 3376$). These results suggested that downregulation of these gene products is likely responsible for the cardiac contraction defect in $Rpl3l^{-/-}$ mice.

A similar analysis was performed for nuclear genes that encode components of the oxidative phosphorylation complexes in the inner mitochondrial membrane that are responsible for ATP production ($n = 91$). The protein subunits of these complexes are encoded by the

nuclear and mitochondrial genomes, with the corresponding transcripts being translated by cytoplasmic ribosomes (including RPL3L-ribosomes) and mitochondrial ribosomes, respectively. Many of these genes in the nuclear genome were also associated with delayed translation elongation at Pro/Ala codons or disome peaks (Fig. 8c). Although the encoded proteins did not show a significant decrease in expression in the RPL3L-deficient heart relative to all proteins, they showed a slightly decreased abundance (Fig. 8d).

Collectively, these results suggested that a reduced abundance of proteins related to cardiac muscle contraction, rather than of proteins

**Fig. 4 | Negative correlation of translation elongation dynamics between conditions of RPL3L overexpression or loss. a** RT-qPCR analysis of *Rpl3* and *Rpl3l* mRNA abundance in C2C12 cells stably expressing RPL3 (RPL3 OE) or RPL3L (RPL3L OE) or in those infected with the corresponding empty retrovirus (*n* = 3 biologically independent samples). *Rpl3* transcripts were measured with primers targeting the coding sequence (CDS) or 3′UTR. Data are means ± s.d. **b** Number of RPL3 or RPL3L molecules per 80S ribosome as determined by MRM analysis of the polysome fraction from RPL3 OE or RPL3L OE cells (*n* = 3 biologically independent samples). Data are means±s.d. **c** Relative ribosome occupancy at A-site codons in RPL3L OE and RPL3 OE cells. Data were aggregated according to all codons for each amino acid (*n* = 3 biologically independent samples). **d** Ribosome occupancy changes at A-site codons for each amino acid residue in RPL3L OE or RPL3 OE cells compared

with control cells (*n* = 3 biologically independent samples). **e, f** Correlation of fold change in ribosome occupancy at A-, P-, and E-site codons for RPL3L OE or RPL3 OE cells (relative to control cells) (*n* = 3 biologically independent samples) with that for the heart of *Rpl3l⁻/⁻* mice (relative to that of control mice) (*n* = 4 mice). Spearman's correlation coefficient (*r*) with associated *P* value (two-sided) is indicated for each comparison. Adjusted *P* values were calculated by the Bonferroni method for multiple comparisons. Gray bands indicate 95% confidence intervals. **g** Violin plots for the CV of ribosome occupancy at the A-site in RPL3L OE, RPL3 OE, and control cells (*n* = 3 biologically independent samples). The inner boxes represent the median and upper and lower quartiles, and the whiskers represent the maximum and minimum values. *$P < 0.05$, ***$P < 0.001$, ns (two-tailed Dunnett's test (**a**) or Steel-Dwass test (**g**)). Source data are provided as a Source Data file.

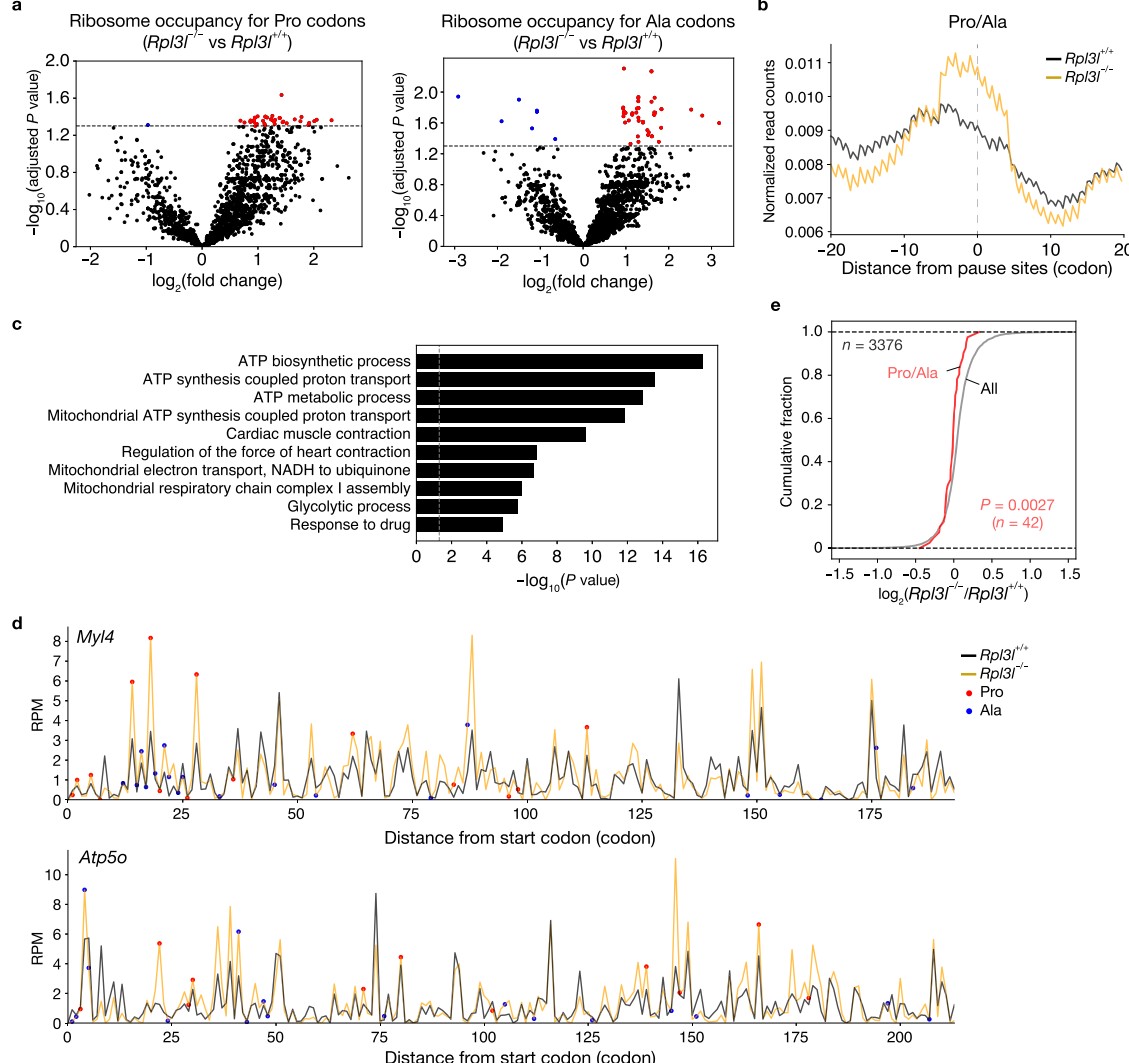

**Fig. 5 | Decreased protein expression associated with aberrant translation elongation dynamics in the RPL3L-ribosome–deficient heart. a** Volcano plots of differential ribosome occupancy for Pro and Ala codons in the heart of *Rpl3l⁻/⁻* mice compared with that of control mice (*n* = 4 mice). *P* values were calculated by two-sided Student's *t* test and adjusted by Benjamini-Hochberg method for multiple comparisons, and the gray dashed lines indicate an adjusted *P* value of 0.05.
**b** Metagene plots for the regions surrounding Pro/Ala sites with altered elongation dynamics for footprints of Ribo-seq analysis performed with the heart of *Rpl3l⁺/⁺* or *Rpl3l⁻/⁻* mice (*n* = 4 mice). The average of replicates is shown. **c** GO analysis of genes with significantly altered elongation dynamics at Pro or Ala codons. *P* values were

calculated as EASE scores, modified Fisher's exact *P* values. The 10 GO terms with the smallest *P* values are listed. **d** Distribution of ribosome footprint occupancy at the A-site along the coding regions of *Myl4* and *AtpSo* mRNAs in the heart of *Rpl3l⁺/⁺* and *Rpl3l⁻/⁻* mice (*n* = 4 mice). The average of four replicates is shown. RPM, reads per million reads. **e** Cumulative fraction for fold change in protein expression in the heart of *Rpl3l⁻/⁻* mice compared with that of control mice (*n* = 4 mice). The expression level of proteins was analyzed by DIA-based MS, and the results are shown for all proteins (*n* = 3376) and for those encoded by the genes with differential ribosome occupancy at Pro or Ala codons. The *P* value was calculated by the two-tailed Mann-Whitney *U* test. Source data are provided as a Source Data file.

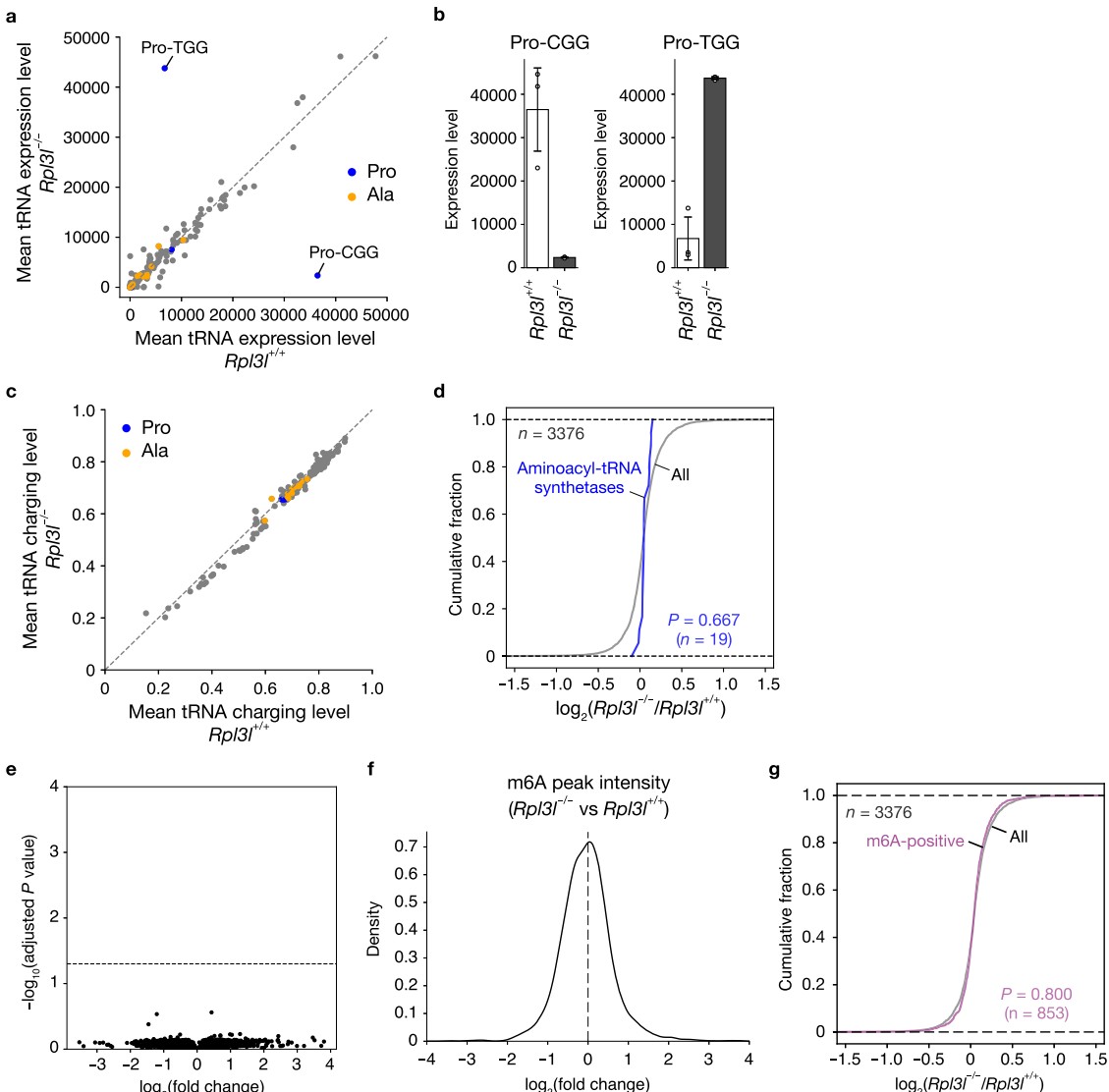

**Fig. 6 | Abundance and aminoacylation status of tRNAs and m6A status of mRNAs in the heart lacking RPL3L. a–c** Mean tRNA abundance, abundance of tRNA-Pro-CGG and tRNA-Pro-TGG, and mean tRNA charging level as determined by Charged DM-tRNA-seq in the heart of $Rpl3l^{+/+}$ and $Rpl3l^{-/-}$ mice ($n = 3$ mice). Data in **b** are means ± s.d. **d** Cumulative fraction for fold change in protein expression as determined by DIA-based MS in the heart of $Rpl3l^{-/-}$ mice compared with that of control mice ($n = 4$ mice). The results are shown for all proteins and aminoacyl-tRNA synthetases. The $P$ value was calculated with the two-tailed Mann-Whitney $U$ test. **e** Volcano plot of differential m6A peak levels as determined by m6A-seq in the heart of $Rpl3l^{-/-}$ mice compared with that of control

mice ($n = 3$ mice). $P$ values were calculated by two-sided Student's $t$ test and adjusted by Benjamini-Hochberg method for multiple comparisons, and the gray dashed line indicates an adjusted $P$ value of 0.05. **f** Histogram for fold change in m6A peak intensity in the heart of $Rpl3l^{-/-}$ mice compared with that of control mice ($n = 3$ mice). The black dotted line corresponds to a fold change of 1. **g** Cumulative fraction for fold change in protein expression as determined by DIA-based MS in the heart of $Rpl3l^{-/-}$ mice compared with that of control mice ($n = 4$ mice). Results are shown for all proteins and for those encoded by genes with m6A peaks. The $P$ value was calculated with the two-tailed Mann-Whitney $U$ test. Source data are provided as a Source Data file.

in the mitochondrial respiratory chain, is likely responsible for the impaired cardiac contractility observed in RPL3L-deficient mice.

## Discussion

We have here characterized the function of RPL3L, a striated muscle−specific RP paralog whose overall structure is similar to that of the canonical, ubiquitously expressed RPL3. We now show that RPL3L-ribosomes contribute to the regulation of translation elongation dynamics, in particular at Ala/Pro codons and at collision-prone sites. RPL3L-ribosomes appear to influence translation of the overall transcriptome, but they especially affect that of transcripts of genes related to cardiac muscle contraction and dilated cardiomyopathy. These properties of RPL3L-ribosomes result in increased production of proteins from these transcripts and thereby play a key role in the

regulation of cardiac function. Of note, although all analyses of the heart were performed with male mice, C2C12 myoblasts are derived from a female mouse, suggesting that the effect of RPL3L-ribosomes on translation elongation is independent of sex.

Recent studies have shown that protein quantity and quality are perturbed not only by defects in translation initiation, but also by aberrant dynamics of translation elongation[42]. For example, ribosome collisions have been found to increase with age and to contribute to age-associated proteostasis defects in *Caenorhabditis elegans* and *Saccharomyces cerevisiae*[57]. In addition, polyglutamine expansion in the protein huntingtin inhibits ribosomal translocation during translation elongation, leading to increased ribosome collisions and decreased protein synthesis[58]. Furthermore, RPL39L, which is specifically expressed in sperm, was recently shown to contribute to cell type-

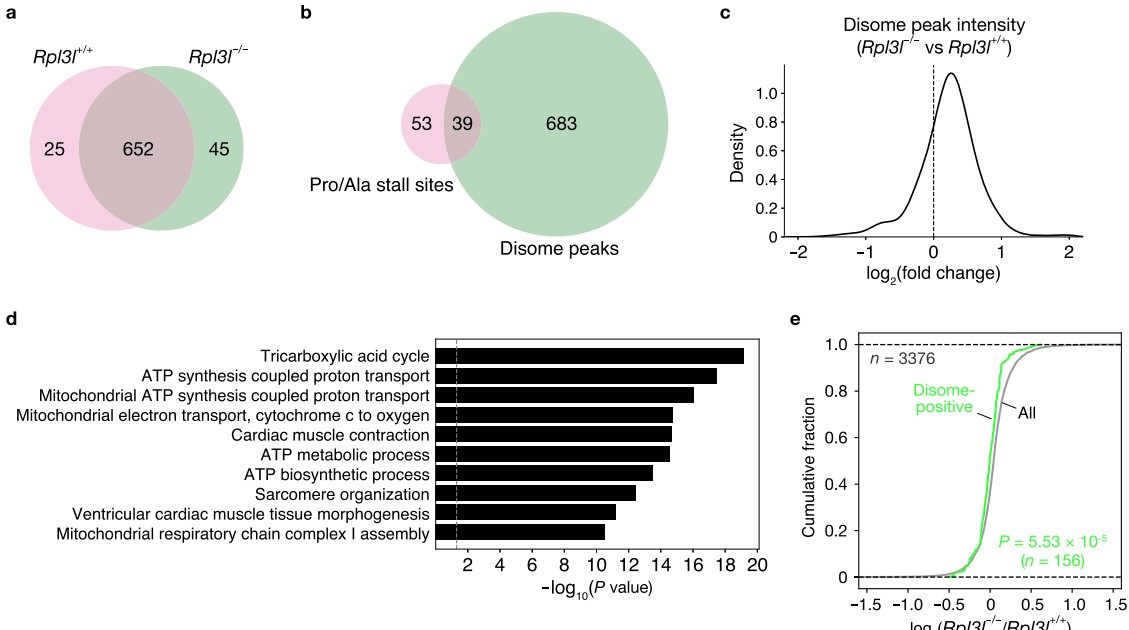

**Fig. 7 | Increased ribosome collision in the RPL3L-deficient heart. a, b** Venn diagrams for the number of disome peaks in the heart of $Rpl3l^{+/+}$ or $Rpl3l^{-/-}$ mice ($n = 4$ mice) (**a**) as well as for the number of Pro/Ala codons with significantly increased ribosome occupancy at the A-site in the heart of $Rpl3l^{-/-}$ mice compared with that of control mice (**b**). **c** Histogram for fold change in disome peak intensity in the heart of $Rpl3l^{-/-}$ mice compared with that of control mice ($n = 4$ mice). The black dotted line corresponds to a fold change of 1. **d** GO analysis of genes with disome peaks in the heart of $Rpl3l^{+/+}$ or $Rpl3l^{-/-}$ mice. $P$ values were calculated as EASE scores, modified Fisher's exact $P$ values. The 10 GO terms with the smallest $P$ values are listed. **e** Cumulative fraction for the fold change in protein expression in the heart of $Rpl3l^{-/-}$ mice compared with that of control mice ($n = 4$ mice). The expression level of proteins was analyzed by DIA-based MS, and the results are shown for all proteins ($n = 3376$) and for those with disome peaks ($n = 156$) in **a**. The $P$ value was calculated with the two-tailed Mann-Whitney $U$ test. Source data are provided as a Source Data file.

specific regulation of translation elongation[59]. RPL39L is located in the exit tunnel for nascent proteins, and the exit tunnel of RPL39L-containing ribosomes differs from that of RPL39-containing ribosomes in both size and charge state. RPL39L-containing ribosomes influence the cotranslational folding of a subset of male germ cell-specific proteins essential for spermatogenesis, and ablation of RPL39L resulted in increased protein aggregation associated with the abnormal folding of such proteins, rendering male mice infertile. These various observations indicate that translation elongation is key to maintenance of protein quantity and quality. The present study of RPL3L-ribosomes also shows the physiological importance of the regulation of translation elongation. However, given the importance of translation initiation to the global translation level, it is possible that the loss of RPL3L also affects translation initiation by some unknown mechanism, a possibility that needs to be examined in future studies.

The NH$_2$-terminal extension, W finger, and basic thumb of RPL3 and RPL3L are located close to the A-tRNA binding pocket of the PTC. In *S. cerevisiae*, mutations in these regions of Rpl3 affect peptidyl transferase activity[27]. Crystallographic analysis has shown that the W255C mutation disrupts the 25S rRNA region comprising nucleotides A2872 to U2875, which forms the A-tRNA binding pocket of the PTC[26]. Whereas increased flexibility of this region conferred by this mutation would be expected to facilitate aminoacyl-tRNA entry into the PTC, it also reduces the precision of amino acid positioning at the A-site, resulting in a reduced efficiency of peptidyl transfer. In addition, His[243] of Rpl3 is methylated by Hpm1 in *S. cerevisiae*, and loss of this methylation affects translation elongation and fidelity[60,61]. METTL18, an ortholog of Hpm1, methylates His[245] (the equivalent residue of His[243] in yeast Rpl3) of RPL3 in humans[28]. This methylation delays translation elongation at Tyr codons, with this delay being required for the proper folding of nascent proteins, and defective methylation results in an increased rate of elongation and in aggregation of Tyr-rich proteins[29].

Given that RPL3L differs from RPL3 at some amino acid positions near the A-tRNA binding pocket and that these amino acid differences are evolutionarily conserved among RPL3L proteins, the change in translation elongation dynamics induced by RPL3L loss may be attributable to structural differences around the A-tRNA binding pocket between RPL3L-ribosomes and canonical ribosomes. In addition, given the difference in elongation efficiency depending on the incorporated amino acid in the RPL3L-deficient heart, RPL3L may also affect the tunnel entrance for the nascent peptide. Further detailed biochemical and structural analyses are warranted to characterize the actual structure of RPL3L-ribosomes and to determine how such structural differences affect translation elongation dynamics.

A recent study of RPL3L-deficient mice by Milenkovic and coauthors have reported results that differ substantially from ours[20]. This previous study thus did not reveal impaired cardiac contractility in the mutant mice. We analyzed the echocardiographic profiles of 18 to 21-week-old mice as well as of 20 to 27-week-old mice in the case of the pressure overload experiments, whereas 8-week-old mice were subjected to such analysis in the previous study, suggesting that a difference in age may explain the difference in results. In addition, Grimes and coauthors have recently generated RPL3L-deficient mice and found that the heart of 18-month-old mutant mice is significantly smaller than that of controls without apparent defects in cardiac contraction[62]. The C57BL/6NTac substrain was used in the analysis by Grimes et al., whereas the C57BL/6J substrain was applied in the analysis by Milenkovic et al. as well as by us, suggesting that this genetic difference may be responsible for the differences in cardiac phenotypes. Milenkovic et al. showed that loss of RPL3L increased ribosome-mitochondria interactions with elevated mitochondrial activity, but Grimes et al. failed to detect any defects in ribosome localization, ultrastructure, or mitochondrial function in the cardiac muscle of RPL3L-deficient mice.

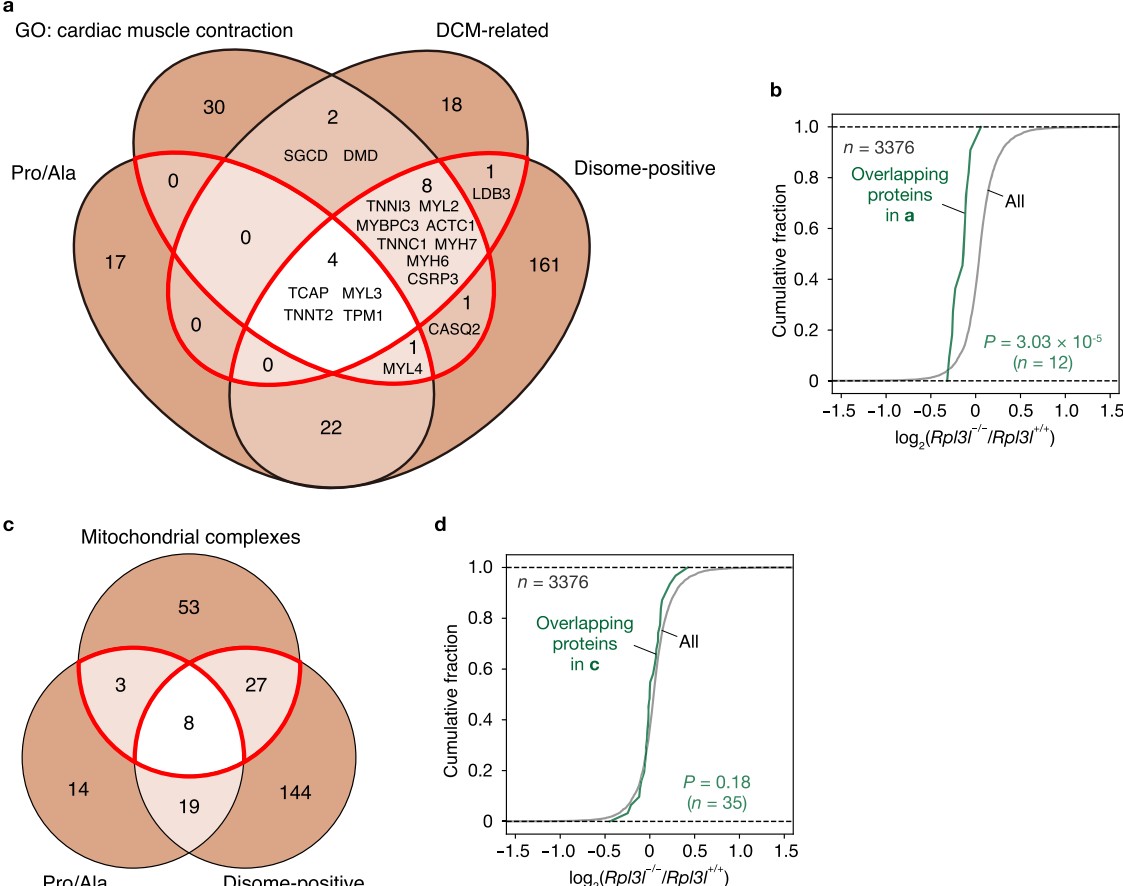

**Fig. 8 | Loss of RPL3L attenuates the expression of proteins related to cardiac muscle contraction. a** Venn diagram showing the number of genes related to cardiac muscle contraction or dilated cardiomyopathy (DCM) as well as of those with differential ribosome occupancy at Pro/Ala codons or with disome peaks in the heart of $Rpl3l^{+/+}$ or $Rpl3l^{-/-}$ mice ($n = 4$ mice). **b** Cumulative fraction for the fold change in protein expression in the heart of $Rpl3l^{-/-}$ mice compared with that of control mice ($n = 4$ mice). The expression level of proteins was analyzed by DIA-based MS, and the results are shown for all proteins ($n = 3376$) and for those encoded by the overlapping genes enclosed by the red boundaries in **a** ($n = 12$). The $P$ value was calculated with the two-tailed Mann-Whitney $U$ test. **c** Venn diagram

showing the number of genes for mitochondrial respiratory complex subunits encoded by the nuclear genome as well as of those with differential ribosome occupancy at Pro/Ala codons or with disome peaks in the heart of $Rpl3l^{+/+}$ or $Rpl3l^{-/-}$ mice ($n = 4$ mice). **d** Cumulative fraction for the fold change in protein expression in the heart of $Rpl3l^{-/-}$ mice compared with that of control mice ($n = 4$ mice). The expression level of proteins was analyzed by DIA-based MS, and the results are shown for all proteins ($n = 3376$) and for those encoded by the overlapping genes enclosed by the red boundaries in **c** ($n = 35$). The $P$ value was calculated with the two-tailed Mann-Whitney $U$ test. Source data are provided as a Source Data file.

Milenkovic and coauthors also performed Ribo-seq analysis for the heart of 8-week-old RPL3L-deficient mice without finding a difference in ribosome occupancy at the A-site, whereas we did find such a difference in the heart of RPL3L-deficient mice at 10 weeks of age. The age difference between the two studies in this case was negligible, leaving the reason for the difference in the results unclear. Although ribosomes are highly sensitive to stress during tissue extraction and high-quality data are required for ribosome occupancy analysis, our Ribo-seq data showed high 3-nt periodicity. Delayed translation elongation at Pro/Ala codons in the RPL3L-deficient heart was also reproducibly observed with both our algorithm and the RUST algorithm, and the translation dynamics in C2C12 cells stably expressing RPL3L were highly inversely correlated with those in the RPL3L-deficient heart, whereas no substantial change in ribosome occupancy at Pro/Ala codons was apparent in C2C12 cells overexpressing RPL3. Moreover, we detected reduced protein levels in the RPL3L-deficient heart for genes with delayed elongation at Pro/Ala codons and for genes prone to ribosome collisions. This consistency of our results is indicative of their reliability.

Although mutations in the *RPL3L* gene have been identified in patients with atrial fibrillation or pediatric dilated

cardiomyopathy[21-25], our analysis of RPL3L-deficient mice did not reveal such pronounced abnormalities even under the pressure overload condition, possibly because of other species differences or interactions with other genes. Elucidation of the specific functions of RPL3L-ribosomes may lead to the development of therapeutic interventions for such diseases.

## Methods
### Animals
All animal experiments were approved by the animal ethics committee of Kyushu University (A20-169-0, A21-271-0, and A22-013-0) and were conducted in compliance with the university guidelines and regulations for animal experimentation. For generation of $Rpl3^{-/-}$ and $Rpl3l^{-/-}$ mice, ribonucleoprotein was prepared by mixing the CRISPR RNA (crRNA) and transactivating crRNA (tracrRNA) with recombinant Cas9 protein (Integrated DNA Technologies) (Supplementary Data 9). Mouse zygotes of the C57BL/6 J strain were subjected to electroporation with each ribonucleoprotein complex. All mice were housed in the specific pathogen-free animal facility at Kyushu University in accordance with institutional guidelines under the following conditions: ambient temperature of 22 °C, 50% to 60% humidity, 12 h-dark/12 h-

light cycle, and free access to water and rodent chow CA-1 (CLEA Japan). Male mice were used for all experiments, given that male mice are commonly used for cardiac analysis. Mice were euthanized in a $CO_2$ chamber for experiments.

## Cell lines and culture conditions

HEK293T (CRL-11268) and C2C12 (CRL-1772) cells were obtained from American Type Culture Collection and were checked for mycoplasma contamination with the use of MycoAlert (Lonza). The cells were cultured under an atmosphere of 5% $CO_2$ at 37 °C in Dulbecco's modified Eagle's medium (DMEM) supplemented with 10% fetal bovine serum (Life Technologies) and antibiotics.

## Retrovirus expression system

Mouse RPL3 and RPL3L cDNAs were subcloned into pCX4/puro (kindly provided by T. Akagi), and the resulting constructs together with pGP (Takara) and pE-eco (Takara) were introduced into HEK293T cells with the use of the FuGENE6 reagent (Roche). Culture supernatants containing recombinant ecotropic retroviruses were harvested and used to infect C2C12 cells by incubation for 24 h in the presence of polybrene (2 μg/ml). The cells were then subjected to selection for 48 h in medium containing puromycin (2.5 μg/ml) and were maintained at low confluence.

## RT-qPCR analysis

Total RNA was extracted from tissue or cell lysates with the use of a PureLink RNA Mini Kit (Thermo Fisher Scientific) and was subjected to RT with the use of a QuantiTect Reverse Transcription Kit (Qiagen). The resulting cDNA was subjected to real-time PCR analysis with specific primers (Supplementary Data 9) and with the use of Luna Universal qPCR Master Mix (New England BioLabs) and a StepOnePlus Real-Time system (Applied Biosystems). The amount of target mRNAs was normalized by that of β-actin mRNA.

## Luciferase reporter assay

A DNA sequence encoding the promoter region of mouse *Rpl3* (from the transcription start site to 1.2 kbp upstream) was cloned into the pGL2-basic vector (Promega), and the resulting construct together with pRL CMV (Promega) was introduced into C2C12 cells with the use of the FuGENE HD Transfection Reagent (Promega). After transfection for 2 days, luciferase activities were assayed with the use of a Dual-Luciferase Reporter Assay System (Promega) and Lumat LB 9510 Tube Luminometer (Berthold Technologies).

## Induction of myoblast differentiation into myotubes

C2C12 cells were transferred to 24-well plates and cultured in DMEM supplemented with 10% fetal bovine serum to full confluence. The medium was then replaced with DMEM supplemented with 2% horse serum, and the cells were cultured for 4 days to induce myoblast to myotube differentiation. The cells were fixed with 4% formalin, incubated for 1 h at room temperature with phosphate-buffered saline (PBS) containing 5% bovine serum albumin (BSA) and 0.2% Triton X-100, and then incubated overnight at 4 °C with a monoclonal antibody to myosin (skeletal, fast) (clone MY-32, M7246, Sigma, 1:250 dilution) in IF buffer (PBS containing 1% BSA and 0.3% Triton X-100). They were washed three times with PBS, incubated for 1 h at room temperature with Alexa Fluor 488–conjugated secondary antibodies (A-11029, Thermo Fisher Scientific, 1:1000 dilution) in IF buffer, and mounted with the use of Vectashield Hard Set Mounting Medium with DAPI (Vector Laboratories) for observation with a Zeiss LSM700 microscope. Quantification of myotube diameter and myogenic fusion was performed with the use of ImageJ. The fusion index was calculated as the number of nuclei in myosin-positive myotubes divided by the total number of nuclei.

## Histopathologic analysis

Tissue was fixed with 4% paraformaldehyde in PBS, embedded in paraffin, and sectioned with a cryostat at a thickness of 3 μm. For analysis of fibrosis, sections were subjected to Masson's trichrome staining and examined with a BZ-X700 microscope (Keyence). For analysis of myofiber size, sections were stained with 5 μg/mL Alexa Fluor 488–conjugated WGA (Thermo Fisher Scientific), mounted with the use of Vectashield Hard Set Mounting Medium with DAPI (Vector Laboratories), and observed with a BZ-X700 microscope.

## Echocardiography

Mice at 18 to 21 weeks of age were subjected to light anesthesia with 1% to 2% isoflurane before echocardiography. For pressure overload experiments, Ang II (1.44 mg/kg per day, Sigma Aldrich) or saline was continuously infused into mice at 20 to 27 weeks of age with the use of a micro–osmotic pump (model 1007D, Alzet) for 1 week. Two-dimensional targeted M-mode images were obtained from the short-axis view at the papillary muscle level with a Vevo 2100 ultrasonography system (Visual Sonics). LVEF was calculated according to the formula: LVEF=[(diastolic LV volume − systolic LV volume)/diastolic LV volume] ×100, where LV volume=[7.0/(2.4 + LV diameter)] × (LV diameter)$^3$. LVWT was calculated as the average of interventricular septum thickness and posterior wall thickness. E, A, the E/A ratio, and deceleration time (DcT) were determined by measurement of flow velocities across the mitral valve by pulsed Doppler imaging. Tissue Doppler was used to measure the mitral annular plane velocity (E' and A'). After echocardiography, mice were euthanized and their tissues weighed.

## Measurement of blood pressure and heart rate

Blood pressure and heart rate were measured in mice with a non-invasive tail-cuff system (model MK-1030 NIBP Monitor for Rats & Mice) as previously described[63].

## Structure prediction

Models of the human 80S ribosome (PDB ID: 6IP5) and A-tRNA (PDB ID: 4V5D) were obtained from the Protein Data Bank[34]. The RPL3-highlighted graphical representation (Fig. 1f) was produced with the use of UCSF ChimeraX software[64]. The predicted model of human RPL3L was obtained from the AlphaFold protein structure database (Identifier: AF-Q92910-F1) and was superimposed onto RPL3 in the 80S ribosome with the use of UCSF ChimeraX.

## Sample preparation for DIA and MRM analysis as well as for Ribo-seq and Disome-seq

C2C12 cells were lysed with a lysis buffer (50 mM Tris-HCl (pH 7.5), 150 mM NaCl, 5 mM $MgCl_2$, 1 mM dithiothreitol, 1% Triton X-100, and EDTA-free protease inhibitor cocktail (Sigma Aldrich)) supplemented with cycloheximide (100 μg/ml). The heart of $Rpl3l^{-/-}$ or $Rpl3l^{+/+}$ mice at 10 weeks of age was flash frozen in liquid nitrogen and lysed in the same solution with the use of a Multi Beads Shocker device (Yasui Kikai). The cell and tissue lysates were centrifuged at $20,380 \times g$ for 10 min at 4 °C, and the resulting supernatants were collected for DIA-based MS analysis as well as for Ribo-seq and Disome-seq, or were loaded on a 1 M sucrose cushion (containing 20 mM Tris-HCl (pH 7.5), 150 mM NaCl, 5 mM $MgCl_2$, 1 mM dithiothreitol, and cycloheximide (100 μg/ml)) for preparation of a polysome fraction. The gradient was centrifuged at 100,000 rpm ($417,200 \times g$) for 1 h at 4 °C in a Beckman TLA110 rotor, and the resulting pellet (polysome fraction) was suspended in Triton lysis buffer (50 mM Tris-HCl (pH 7.5), 150 mM NaCl, 0.5% Triton X-100, 4 mM sodium orthovanadate, 4 mM EDTA, 100 mM NaF, 100 mM sodium pyrophosphate, 1 mM phenylmethylsulfonyl fluoride) and subjected to MRM analysis.

## DIA analysis

Protein concentration of the lysate supernatants was adjusted to 1 mg/ml, and cysteine residues were reduced with 2.5 mM tris(2-carboxyethyl)phosphine hydrochloride (Thermo Fisher Scientific) for 30 min at 37 °C and alkylated with 5 mM 2-iodoacetamide (Sigma Aldrich) for 30 min at room temperature. Proteins were precipitated with acetone, dispersed by ultrasonic treatment in 25 mM triethylammonium bicarbonate, and digested with trypsin (Laurus Bio). The resulting peptides were purified with the use of an SDB-XC-StageTip (3 M), injected (200 ng/sample) into a pre-column (L-column2 micro, CERI), and fractionated on an in-house–fabricated 20 cm column packed with 2 μm octadecyl silane particles (CERI). Elution was performed with a linear gradient of 5% to 35% solvent B over 70 min at a flow rate of 200 nl/min (solvent A = 0.1% formic acid; solvent B = 0.1% formic acid and 99.9% acetonitrile) and with the use of a Dionex Ultimate 3000 HPLC System (Thermo Fisher Scientific). Eluted peptides were sprayed with a nano-electrospray source and with a column oven set at 42 °C (AMR). The Q Exactive Hybrid Quadrupole-Orbitrap mass spectrometer (Thermo Fisher Scientific) was operated in DIA mode. All data were acquired in profile mode with positive polarity. MS1 spectra were collected in the mass/charge ($m/z$) ratio range of 430 to 860 at a resolution of 35,000 with an automated gain control (AGC) target value of $1 \times 10^6$ and a maximum injection time of 50 ms. MS2 spectra were collected in the $m/z$ range of >200 at a resolution of 17,500 with an AGC target value of $1 \times 10^6$ and with the automatic maximum ion injection time. Twenty-one DIA windows of 20 units were set from an $m/z$ of 430 to 850 with an overlap of 1 unit. Normalized collision energy was set to 25%. All DIA raw data were processed with DIA-NN (version 1.8.1) in the library-free search mode with reference to mouse UniProt sequences with the addition of RPL3L (UniProt accession: E9PWZ3)[65].

## MRM analysis

For production of internal standard peptides for MRM analysis, pENTR2B vectors (Thermo Fisher Scientific) containing target peptide sequences, which were concatenated into several synthetic artificial protein sequences, were produced by Eurofins Genomics (Supplementary Data 10) and were then subjected to recombination with the pET21b-QcodeAA-DEST vector with the use of Gateway LR Clonase II Enzyme Mix (Thermo Fisher Scientific). Recombinant proteins were expressed in an *Escherichia coli* expression strain that was auxotrophic for arginine and lysine and grown in the presence of $[^{13}C_6/^{15}N_4]$Arg and $[^{13}C_6/^{15}N_2]$Lys, and were subsequently purified with the use of Ni-resin (Probond, Invitrogen). The protein concentration of the samples and internal standard peptides was determined with the BCA assay (Bio-Rad). Cysteine residues were blocked by incubation of the samples with 2 mM tris(2-carboxyethyl)phosphine hydrochloride for 30 min at 37 °C and then alkylated with 10 mM 2-iodoacetamide for 30 min at room temperature. The samples were then digested with trypsin (Thermo Fisher Scientific) at 5 μg/ml for 14 h at 37 °C. MRM analysis was performed with a QTRAP5500 instrument (SCIEX) operated in positive-ion mode. Typical parameters were set as follows: spray voltage of 2300 V, curtain gas setting of 10, collision gas setting of high, ion source gas 1 setting of 30, and interface heater temperature of 160 °C. Collision energy (CE) was calculated with the following formulas: $CE = (0.044 \times m/z1) + 5.5$ and $CE = (0.051 \times m/z1) + 0.5$ for doubly and triply charged precursor ions, respectively, and where $m/z1$ is the mass/charge ratio for the precursor ion. The collision cell exit potential (CXP) was calculated according to the formula: $CXP = (0.0391 \times m/z2) - 2.2334$, where $m/z2$ is the mass/charge ratio for the fragment ion. The declustering potential (DP) was set to 50, and the entrance potential (EP) was set to 10. Resolution for Q1 and Q3 was set to "unit" (half-maximal peak width of 0.7 Da). The scheduled MRM option was adopted for all data acquisition, with a target scan time of 6.7 s and MRM detection windows of 300 s for verification of MRM assays. Raw data were analyzed by iMPAQT-Quant with the

corresponding spectral library. Peak groups were scored on the basis of cosine similarity with the MS/MS spectra obtained in data-dependent acquisition (DDA) mode, a peak coelution of at least three fragment ions for each peptide, the presence or absence of interfering ions, and intensity. Finally, all traces were manually checked to eliminate inadequate transitions. Artificial protein sequences and corresponding MRM assays are listed in Supplementary Data 11.

## Ribo-seq and Disome-seq

Cell and tissue lysate supernatants were assayed for RNA concentration with a Qubit RNA BR Assay Kit (Thermo Fisher Scientific), incubated with RNase I (20 U per 10 μg of RNA, Epicentre) for 45 min at 25 °C, and placed on ice before the addition of SUPERase•In RNase inhibitor (Invitrogen) to a concentration of 20 U/ml. The samples were then loaded on a 1 M sucrose cushion (containing 20 mM Tris-HCl (pH 7.5), 150 mM NaCl, 5 mM MgCl$_2$, SUPERase•In (10 U/ml), 1 mM dithiothreitol, and cycloheximide (100 μg/ml)), and the gradients were centrifuged at 100,000 rpm ($417,200 \times g$) for 1 h at 4 °C in a Beckman TLA110 rotor. The resulting pellets were suspended in ribosome splitting buffer (20 mM Tris-HCl (pH 7.5), 300 mM NaCl, 5 mM EDTA, SUPERase•In (10 U/ml), 1 mM dithiothreitol, 1% Triton X-100) and subjected to purification with the use of an Amicon Ultra filtration device (100-kDa cutoff, Millipore) to deplete rRNAs. The purified RNA was subjected to selection on the basis of a size range of 17 to 34 nt for Ribo-seq and 50 to 80 nt for Disome-seq by electrophoresis through a 15% polyacrylamide and Tris-borate-EDTA (TBE)−urea gel (SuperSep RNA, Fujifilm). The footprint fragments were treated with T4 polynucleotide kinase (New England BioLabs) to repair 2′−3′ cyclic phosphates, and a DNA linker including barcode sequences (NI-810 to NI-817) (Supplementary Data 9) was ligated with the use of T4 RNA ligase 2, truncated K227Q (New England BioLabs). The resulting products were purified on a 15% polyacrylamide and TBE-urea gel, and rRNAs were further depleted with the use of RiboZero Gold (Illumina). RT was performed with the NI-802 primer (Supplementary Data 9), and the resulting products were purified on a 15% polyacrylamide and TBE-urea gel. The purified cDNAs were circularized with circLigase II (Lucigen), and index sequences were then added by PCR amplification with the common primer NI-798 and primers including index sequences (NI-799 and NI-822 to NI-826).(Supplementary Data 9). Products of the desired size were purified on a 15% polyacrylamide nondenaturing gel (SuperSep DNA, Fujifilm), and the libraries were sequenced with a NovaSeq 6000 system (Illumina).

## RNA-seq

Total RNA was extracted from tissue or cell lysates with the use of a PureLink RNA Mini Kit (Thermo Fisher Scientific), and the quality of the purified RNA was assessed with a 2100 Bioanalyzer (Agilent). After mRNA selection with the use of a NEBNext Poly(A) mRNA Magnetic Isolation Module (New England BioLabs), libraries were prepared with the use of a NEBNext Ultra Directional RNA Library Prep Kit for Illumina (New England BioLabs). The cDNAs were sequenced with a NovaSeq 6000 system.

## Charged DM-tRNA-seq

Charged DM-tRNA-seq was performed as previously described[45]. Periodic oxidation of 10 μg of total RNA extracted from mouse heart was performed for 30 min at room temperature in a solution containing 100 mM acetic acid/sodium acetate buffer (pH 4.8) and 50 mM NaIO$_4$. Glucose (1 M solution) was then added to a final concentration of 100 mM, the mixture was incubated for 5 min at room temperature, and RNA was purified with the use of a Micro Bio-Spin P-6 Column (Bio-Rad). The RNA was treated with 60 mM sodium borate (pH 9.5) for 90 min at 45 °C for deacylation and purified again with a Micro Bio-Spin P-6 Column. The purified RNA was selected according to size in the range of 70 to 90 nt by electrophoresis

through a 15% polyacrylamide and TBE-urea gel, and then demethylated with the demethylase supplied with an rtStar tRNA-Optimized First-Strand cDNA Synthesis Kit (Filgen). The resulting sample was subjected to dephosphorylation, DNA linker ligation, and rRNA depletion with RiboZero Gold as described for the Ribo-seq protocol. RT was performed with the SI-019 primer, and the resulting products were purified on a 15% polyacrylamide and TBE-urea gel. The second linker (SI-018) was ligated by incubation overnight at 25 °C with T4 RNA Ligase 1 (ssRNA Ligase), High Concentration (New England BioLabs). The products were then amplified by PCR with the same primers as used for the Ribo-seq protocol, and those of the desired size were purified by electrophoresis through a 15% polyacrylamide nondenaturing gel. The libraries were sequenced with a NovaSeq 6000 system (Illumina).

### m6A-seq
m6A-seq libraries were prepared with the use of an EpiNext CUT&RUN RNA m6A-Seq Kit (EpiGentek). In brief, total RNA (5 μg) extracted from mouse heart was subjected to immunoprecipitation with the m6A antibody (P-9016, EpiGentek, 1:100 dilution) and cleaved on beads. The beads were then washed, RNA was purified from the beads and subjected to RT, and the resulting cDNA was amplified by PCR. The libraries were sequenced with a NovaSeq 6000 system (Illumina).

### Read processing and analysis for Ribo-seq and Disome-seq
Adaptor sequences were trimmed from raw reads with the use of cutadapt. Reads of low quality were discarded with the use of fastq_quality_trimmer and fastq_quality_filter of the FASTX-Toolkit. Ribosomal RNA reads were removed by alignment with mouse rRNA sequences with the use of STAR, and the remaining reads were aligned with the mouse transcriptome (GRCm38.p6) and mouse genome (mm10) also with the use of STAR. Multiple mappings were allowed. Aligned reads were sorted and indexed with the use of Samtools. UMI deduplication was performed with the use of UMI-tools. The A-site was defined as the position located 15 nt from the start of reads for 28- to 30-nt reads that showed 3-nt periodicity. Three-nucleotide periodicity was analyzed with RiboCode and RibORF, and footprints with a score of ≥0.5 were applied to the analysis. Although Ribo-seq libraries were prepared with footprints of 17 to 34 nt, given that footprints derived from ribosomes with open A-sites are 20 to 22 nt in length[66], 20- to 22-nt footprints were excluded from the analysis because most of them failed to show 3-nt periodicity in the libraries generated in this study. Reads mapped to the mouse transcriptome were counted with the use of featureCounts (Subread package). The first and last five codons were excluded from the analysis to avoid the atypical footprint counts observed around start and stop codons[67,68]. Ribosome occupancy was calculated as the average of the normalized read counts (RPM) corresponding to the positions of the E-, P-, and A-sites, and was further corrected by the average of the total codon RPMs to obtain the relative ribosome occupancy. Codon-specific dwell times were also calculated with the use of RUST[41].

### Read processing and analysis for charged DM-tRNA-seq
Adaptor sequences were trimmed from raw reads with the use of cutadapt. The remaining reads were aligned to the reference tRNA sequences obtained from GtRNAdb with the use of the python script "Reference Sequence Alignment.py," and read count and aminoacylation ratios were determined with the use of the python script "creating output from sequence_alignments.py." These python scripts were downloaded from https://github.com/Jessica-Pan/DM-tRNA-seq-ref-genomes.

### Read processing and analysis for m6A-seq
Adaptor trimming, quality filtering, depletion of rRNA reads, and read mapping were performed as described for Ribo-seq and Disome-seq.

Multiple mappings were allowed. Peak calling was performed with the use of HOMER findPeaks. The number of reads was determined with the use of bedtools for the peak on the coding region commonly identified in three replicates for each of the $Rpl3l^{+/+}$ and $Rpl3l^{-/-}$ samples.

### scRNA-seq analysis
Analysis of published scRNA-seq data was performed with the use of the R package Seurat or Scanpy (v1.9.1). For analysis of scRNA-seq data from human heart, the processed R object file was downloaded from GSE183852. For analysis of scRNA-seq data from mouse heart, the gene expression matrix data were downloaded from Tabula Muris.

### Statistical analysis
DESeq2 was applied to analyze differential expression based on RNA-seq and Ribo-seq reads[69], and the generalized linear model in RiboDiff was used to analyze differential TE[70]. GO analysis and EASE score calculation were performed by DAVID. Spearman's correlation with associated $P$ value was calculated in Python 3.6.8 with the use of Pandas (v0.25.3). Student's $t$ test and the Mann-Whitney $U$ test were performed in Python 3.6.8 with the use of Scipy (v1.2.1). The Steel-Dwass test was performed in scikit-posthocs (v0.6.7). Welch's $t$ test was performed with the use of Microsoft Excel, and Dunnett's test and the Tukey-Kramer test with JMP16. A $P$ value of <0.05 was considered statistically significant. For extraction of differentially expressed genes, we used an adjusted $P$ value (false discovery rate (FDR) $q$ value) of <0.05.

### Reporting summary
Further information on research design is available in the Nature Portfolio Reporting Summary linked to this article.

## Data availability
All sequence data have been deposited in GEO under the accession number GSE203072. The MS data have been deposited with the ProteomeXchange Consortium via the JPOST partner repository under the data set identifiers PXD034018 and PXD034019. Structural data for human 80S ribosome (PDB ID: 6IP5 and A-tRNA (PDB ID: 4V5D were obtained from the Protein Data Bank. The predicted model of human RPL3L was obtained from the AlphaFold protein structure database (Identifier: AF-Q92910-F1). Source data are provided with this paper.

## Code availability
All custom Python scripts used in the analyses of this paper are available on request.

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

## Acknowledgements

We thank L. Cui and the Research Promotion Unit of the Medical Institute of Bioregulation at Kyushu University for technical assistance, C. Kikutake for advice on statistical analysis, as well as A. Ohta for help with preparation of the manuscript. Computations were performed in part on the NIG supercomputer at ROIS National Institute of Genetics. This research was supported in part by KAKENHI grants from Japan Society for the Promotion of Science (JSPS) and the Ministry of Education, Culture, Sports, Science, and Technology of Japan to A.M. (JP20H05928) and to K.I.N. (JP18H05215), as well as by a grant from the Japan Agency for Medical Research and Development (21wm0425002) to K.I.N.

## Author contributions

C.S. performed experiments and wrote the manuscript. A.M. conceived and designed the project, performed experiments, and wrote the manuscript. K.I. performed computational analysis and wrote the manuscript. T.Ya. performed echocardiography experiments. T.Y. performed structure analysis. T.M. performed library preparation. A.H. and M.M. performed MS experiments. Y.T., E.M.-S., S.I., S.M., and H.T. supervised experimental design. K.I.N. coordinated the study and wrote the manuscript.

## Competing interests

The authors declare no competing interests.
