## [Peer Review File · Nature Communications]

RPL3L-containing ribosomes determine translation elongation dynamics required for cardiac functionREVIEWER COMMENTS

Reviewer #1 (Remarks to the Author):

In this manuscript Shirishi et al investigate the role of tissue specific ribosomal protein RPL3L (paralog of RPL3) and provide evidence for its role in cardiac function. Furthermore, the authors suggest that this role is mediated via ribosomes containing RPL3L instead of RPL3 to which they refer as myoribosomes. They propose that altered properties of these myoribosomes effect translational dynamics. This is certainly a very interesting work and the experimental work is of high quality. However, I would like to challenge the authors interpretations and conclusions which in my opinion are too narrow and biased towards a single plausible hypothesis while ignoring other possible explanations of the observed data. I also have a few technical comments.

General comments on interpretation.

I find the evidence for RPL3L involvement in cardiac function to be strong (the caveat is that I am not an expert in cardiac function). However, I am far less convinced that this involvement is mediated through specialised myoribosomes. It seems to me that the authors strongly favour specialised ribosome idea to explain the data. In my opinion the same observations would be obtained as a result of moonlighting activities of RPL3L. Based on the data presented in this manuscript I couldn't favour either of the hypotheses and I believe that the manuscript should be more balanced in this regard.

The main arguments in the support of specialised ribosome hypothesis are that RPL3L $-/-$ is associated with altered decoding rates of specific codons (slowed at Proline and Alanine, increased at Lys) and reduced ribosome collisions. Further argument is that affected genes are enriched in functions associated with ATP synthesis and cardiac muscle contraction.

The first question is whether these changes in codon elongation rates are responsible for the changes in expression of these genes? RPL3L is likely a cause of both, but I am not convinced that altered elongation rates are the cause of specific changes in gene expression which may be modulated via different mechanisms. My argument against this is that the changes in elongation rates seem marginal and protein synthesis is believed to be largely regulated at the level of initiation. So it is plausible that RPL3L may affect expression of these genes not via altered pauses at specific codons, but through other mechanisms.

Second, even if altered codon decoding rates is indeed the cause of altered gene expression, I am still not convinced that alterations in codon specific rates are caused by RPL3L containing ribosomes. While theoretically possible, despite proximity of RPL3L to PTC, I do not see a straightforward mechanism for how a ribosomal protein could alter decoding of specific codons and not aware of such examples. The alterations of decoding rates are usually caused by changes in the concentrations of corresponding cognate tRNAs, their aminoacylation or their epitranscriptomics status. Thus, to me, it is easier to imagine that RPL3L cause these changes by acting on tRNAs, aminoacyl tRNA synthetases, RNA modification enzymes or some factors regulating their expression or activities and not via changes in the ribosome.

It may even be possible that the causality here is reverse to what is proposed by the authors. It is possible that changes in gene expression cause changes in codon decoding rates because they alter the demand for tRNAs bearing amino acids enriched or depleted in these genes and subsequently change the availability of those tRNAs.

The moonlighting activity of RPL3L is also supported by the inhibition of its expression in the presence of exogenous RPL3L. Since it is RPL3L RNA and not just RPL3L protein that is being reduced, the inhibition ought to occur at RNA level suggesting that RPRL3L can induce transcriptional changes when outside of the ribosome.

Technical questions/suggestions

How ribosome occupancy was calculated?

Local decoding rates of translation on certain amino acids (Pro/Ala/Lys) is different between Rpl3l⁻/Rpl3l⁻ with Rpl3l⁺/Rpl3l⁺ genotypes, but it makes sense to carry out this analysis for specific codons instead of on groups of synonymous codons. This will help to understand whether the effect is on amino acids, specific codons or tRNA isoacceptors .

Furthermore, the way how the local decoding rates (a.k.a. pauses, dwelling times) are calculated could be very sensitive to the input of very few individual codons at which strong pauses occur at highly expressed mRNAs. This makes it difficult to detect global changes, since any changes in codon decoding rates of lowly expressed mRNAs would have negligible contribution to the signal. Thus, I strongly recommend to use RUST method (O'Connor et al DOI: 10.1038/ncomms12915). RUST greatly reduces the impact of individual locations and thus could help identification of global effects. I believe if the authors observation is genuine, RUST would provide a much stronger signal than what was obtained with the authors' method.

In Ribo-seq data processing, the authors excluded the first and the last five codons from the analysis. This is certainly a good idea, but perhaps a justification should be given for the readers unfamiliar with associated issues.

Methods state that RNA fragments of size between 17 to 34 nt were kept although only 28 and 29nt reads were used with arbitrary 15nt offset. Could this be explained? What reads of other lengths were used for?

The combined total number of disome peaks is given for both RPL3L-deficient and control hearts. But no information on their number and overlap between them is given for RPL3L-deficient and control samples.

Multiple testing correction needs to be applied to the p-values obtained for the comparison shown in Fig. 2i, and for correlation p-values on Fig 4e,f.

Why the plot in Fig. 3b has two blue dots for RPL3L?

On Fig. 5f colours for Pro and Ala are almost indistinguishable.

Pasha Baranov

Reviewer #2 (Remarks to the Author):

NCOMMS-22-27933-T Shiraishi et al "Myo-ribosomes" determine translation elongation dynamics required for cardiac function

In this manuscript Shiraishi et al show that show that RPL3L, a paralog of RPL3, which is specifically expressed in heart and skeletal muscle, influences translation elongation dynamics. Deficiency of RPL3L-containing ribosomes, which the authors designated as "myo-ribosomes" in RPL3L knockout mice resulted in impaired cardiac contractility. The authors went on to examine the effects of deleting RPL3L in the mouse heart on mouse heart function under baseline conditions. They then went on to study the effects of RPL3L deletion in mouse hearts or overexpression in C2C12 myoblasts on translational elongation dynamics, ribosome occupancy and selective effects on translation of a subset of affected mRNAs.

Summary:

While this study represents an interesting area of ribosome function in translation in cardiac

myocytes, there remain some methodological and other experimental issues that require attention and resolution, as described below. Moreover, there have been other publications, notably ref. 20, that do much of the same experiments; this decreases the novelty and impact of the current paper and, since different results were obtained in the ref. 20 from those described here, places Shiraishi et al in a position of describing why their results differ from those in ref. 20. Moreover, in ref 20 there were more conclusions drawn as to how RPL3L deletion might affect mitochondrial function in the heart, and in skeletal muscle as well.

Comments:

1- I don't think the authors should use the term "myo-ribosomes", but instead should use the more accurate term, RPL3L-ribosomes, because there is a possibility that other myocyte specific ribosomal proteins will be discovered and that the ribosomes they are part of might not be the same ribosomes as the RPL3L-containing ribosomes.

2- One of the limitations of the study that the authors need to address is that the ribo-tag and RNA-seq are examining mRNAs from all cell types in the heart, while the deletion affects only a subset of the cell types, i.e. some myocytes.

3- While the structural analysis shown in 1f and 1g are interesting, they really aren't critical to the study and may suggest more than the authors want to infer. For example, based on the overlay of RPL3 and RPL3L, the reader is led to believe that these proteins occupy the same site in the ribosome and I don't think the authors know enough to know whether that is true.

4- The effect of RPL3L deletion on mouse heart function is subtle and mild. It would be important to know whether the functional effect is exacerbated under a typical cardiac stress situation, such as pressure overload. In this regard, a table of the complete echocardiography parameters should be included.

5- While it looks convincing that RPL3L deletion changes translational dynamics, it is unclear whether any of these changes contributes to the contractility phenotype seen in the mice, *in vivo*.

6- Related to the comment above, it would be important for the authors to select specific proteins whose expression was affected by RPL3L deletion and to list them and discuss the possible implications. The GO categories are a start, but drilling down deeper is important for readers to interpret the data.

7- C2C12 myoblasts are not a good model system with which to examine RPL3L overexpression (OE), as the authors have done. In fact, in the Introduction the authors state that "Forced expression of RPL3L in C2C12 myogenic cells inhibits myoblast fusion and myotube growth¹⁹, whereas loss of RPL3L increases ribosome-mitochondria interactions in mouse cardiomyocytes²⁰." The inhibition of myoblast fusion and myotube growth would seem to be antithetical to the activity the authors would like to ascribe to RPL3L.

8- In ref. 20 in the current manuscript there is a more complete description of how RPL3L deletion "modulates the expression of RPL3, which in turn affects ribosomal subcellular localization and, ultimately, mitochondrial activity", which better describes the functional consequences of RPL3L depletion than does the current paper. In fact, the current paper's finding that cardiac function/contractility was affected is not what was seen in the paper described in ref. 20, where the authors stated that "echocardiographic profiles of Rpl3l^{-/-} mice showed no significant differences to those of control Rpl3l^{+/+}."

Reviewer #3 (Remarks to the Author):

In the present paper, Shiraishi and colleagues investigate why in heart and skeletal muscle RPL3L, the paralog of canonical uL3, is overexpressed. Overall, tissue-specific expression of paralogs of ribosomal proteins and how this regulates and fine-tunes gene expression is an enigmatic and important phenomenon.

Here, the authors show that in the heart of wt mice about two thirds of ribosomes are so-called myo-ribosomes and have RPL3L instead of uL3. RPL3L-deficient knock-out mice overexpress uL3 to compensate but suffer from impaired cardiac contractility establishing that the loss of myo-ribosomes has specific physiological consequences. Further analysis using Ribo-seq shows that the lack of myo-ribosomes affects translation elongation dynamics for certain codons (delayed at Pro and Ala codons, increased at Asn and Lys codons) and results in increased ribosome collisions/ disome formation. Altered translation elongation dynamics in turn results in reduced expression of proteins related to mitochondrial ATP synthesis and cardiac muscle contraction.

Accordingly, Shiraishi and colleagues demonstrate that RPL3L-containing myo-ribosomes are important for the efficient expression of specific proteins. Whereas the exact molecular mechanism of RPL3L remains unclear, the paper rationalizes the physiological importance of the ribosomal protein paralog and paves the way for further in-depth biophysical and structural analysis. Overall, the paper is well written. I have only some suggestions to improve the paper.

Specific Points:

1. It may be helpful to highlight the position of amino acids that differ between uL3 and RPL3L (Fig. 1h) also in the structural model depicted in Fig. 1g.
2. The authors write that no significant differences for other RPs were detected (page 6, 1st sentence) and refer to extended Data Fig 2c. What about RPS15?
3. In the discussion, the authors speculate that increased flexibility of the PTC region may lead to a reduced efficiency of peptidyl transfer. However, the loss of myo-ribosomes differentially affects elongation efficiency dependent on the amino acid incorporated. Thus, the very entrance the tunnel for the nascent peptide could be affected as well.

Response to Reviewer #1

In this manuscript Shirishi et al investigate the role of tissue specific ribosomal protein RPL3L (paralog of RPL3) and provide evidence for its role in cardiac function. Furthermore, the authors suggest that this role is mediated via ribosomes containing RPL3L instead of RPL3 to which they refer as myoribosomes. They propose that altered properties of these myoribosomes effect translational dynamics. This is certainly a very interesting work and the experimental work is of high quality. However, I would like to challenge the authors interpretations and conclusions which in my opinion are too narrow and biased towards a single plausible hypothesis while ignoring other possible explanations of the observed data. I also have a few technical comments.

[Response] We thank Dr. Baranov for the careful review of our manuscript and for the statements that “*This is certainly a very interesting work and the experimental work is of high quality.*” We also appreciate the many constructive suggestions that we feel have helped us to greatly improve our manuscript. Our specific responses to the points raised are as follows (this letter contains low-resolution thumbnails for clarity; please refer to the manuscript for high-resolution figures):

General comments on interpretation:

I find the evidence for RPL3L involvement in cardiac function to be strong (the caveat is that I am not an expert in cardiac function). However, I am far less convinced that this involvement is mediated through specialised myoribosomes. It seems to me that the authors strongly favour specialised ribosome idea to explain the data. In my opinion the same observations would be obtained as a result of moonlighting activities of RPL3L. Based on the data presented in this manuscript I couldn't favour either of the hypotheses and I believe that the manuscript should be more balanced in this regard.

The main arguments in the support of specialised ribosome hypothesis are that RPL3L -/- is associated with altered decoding rates of specific codons (slowed at Proline and Alanine, increased at Lys) and reduced ribosome collisions. Further argument is that affected genes are enriched in functions associated with ATP synthesis and cardiac muscle contraction.

The first question is whether these changes in codon elongation rates are responsible for the changes in expression of these genes? RPL3L is likely a cause of both, but I am not convinced that altered elongation rates are the cause of specific changes in gene expression which may be modulated via different mechanisms. My argument against this is that the changes in elongation rates seem marginal and protein synthesis is believed to be largely regulated at the level of initiation. So it is plausible that RPL3L may affect expression of these genes not via altered pauses at specific codons, but through other mechanisms.

[Response] As the reviewer points out, it is possible that the loss of RPL3L may have altered something other than codon elongation rates. To address this point as well as the reviewer's later comments, we performed Charged DM-tRNA-seq to determine the abundance and charge status of tRNAs¹, as well as m6A-seq to examine mRNA methylation status. We also performed a promoter activity assay to evaluate the function of RPL3L outside of the ribosome. As described in more detail in the responses to the comments below, we found that a few tRNAs differed in abundance between the RPL3L-deficient and control heart (**new Fig. 6a–d and Supplementary Data 6**), suggesting that changes in the abundance of such tRNAs may have some effect on translation elongation dynamics.

However, given the lack of changes in the abundance and charge status of most tRNAs as well as in the N⁶-methyladenosine (m6A) status of all mRNAs (**new Fig. 6e–g and Supplementary Data 7**), tRNA and epitranscriptome status are unlikely to be largely responsible for the defects in translation elongation dynamics in the RPL3L-deficient heart.

Recent studies have shown that protein quantity and quality are perturbed not only by defects in translation initiation, but also by aberrant dynamics of translation elongation². The speed of translation elongation is not constant even under normal translation conditions, and ribosomes locally accelerate or decelerate to ensure proper translation. Abnormal delays in elongation cause ribosome collisions, leading to the formation of disomes, which trigger the ribosome-associated quality control (RQC) pathway and consequent degradation of nascent proteins. On the other hand, an abnormal acceleration of elongation prevents proper folding of nascent proteins, resulting in the formation of protein aggregates.

For example, ribosome collisions increase with age and contribute to age-associated proteostasis defects in *Caenorhabditis elegans* and *Saccharomyces cerevisiae*³. In addition, polyglutamine expansion of huntingtin causes Huntington's disease in humans and inhibits ribosomal translocation during translation elongation, leading to increased ribosome collisions and decreased protein synthesis⁴. Furthermore, methylation of RPL3 affects translation dynamics. His²⁴⁵ of RPL3 is methylated by the methyltransferase METTL18, and defective methylation increases the rate of elongation at Tyr codons, resulting in the aggregation of Tyr-rich proteins⁵ RPL39L, which is specifically expressed in sperm, has also recently been shown to contribute to cell type-specific regulation of translation elongation⁶. RPL39L is located in the exit tunnel of nascent proteins, and the exit tunnel of RPL39L-containing ribosomes differs from that of RPL39-containing ribosomes in both size and charge state. RPL39L-containing ribosomes regulate the cotranslational folding of a subset of male germ cell-specific proteins essential for spermatogenesis, and ablation of RPL39L results in increased protein aggregation associated with abnormal folding, rendering male mice infertile. These various observations indicate that translation elongation is a critical factor for maintenance of protein quantity and quality. Our study of RPL3L-containing ribosomes now also demonstrates the physiological importance of the regulation of translation elongation.

However, as the reviewer indicates, we cannot formally rule out the possibility that deficiency of RPL3L might affect translation initiation by some unknown mechanism. We have therefore tried to be more balanced in describing both possibilities in the Discussion section of the revised manuscript (page 13, lines 406-425).

[References]

1. Evans ME, Clark WC, Zheng GQ, Pan T. Determination of tRNA aminoacylation levels by high-throughput sequencing. *Nucleic Acids Res.* **45**, e133 (2017).
2. Stein KC, Frydman J. The stop-and-go traffic regulating protein biogenesis: How translation kinetics controls proteostasis. *J. Biol. Chem.* **294**, 2076-2084 (2019).
3. Stein KC, Morales-Polanco F, van der Lienden J, Rainbolt TK, Frydman J. Ageing exacerbates ribosome pausing to disrupt cotranslational proteostasis. *Nature* **601**, 637-642 (2022).
4. Eshraghi M, *et al.* Mutant Huntingtin stalls ribosomes and represses protein synthesis in a cellular model of Huntington disease. *Nat. Commun.* **12**, 1461 (2021).
5. Matsuura-Suzuki E, *et al.* METTL18-mediated histidine methylation of RPL3 modulates translation elongation for proteostasis maintenance. *Elife* **11**, e72780 (2022).
6. Li H, *et al.* A male germ-cell-specific ribosome controls male fertility. *Nature* **612**, 725-731 (2022).

Second, even if altered codon decoding rates is indeed the cause of altered gene expression, I am still not convinced that alterations in codon specific rates are caused by RPL3L containing ribosomes. While theoretically possible, despite proximity of RPL3L to PTC, I do not see a straightforward mechanism for how a ribosomal protein could alter decoding of specific codons and not aware of such examples. The alterations of decoding rates are usually caused by changes in the concentrations of corresponding cognate tRNAs, their aminoacylation or their epitranscriptomics status. Thus, to me, it is easier to imagine that RPL3L cause these changes by acting on tRNAs, aminoacyl tRNA synthetases, RNA modification enzymes or some factors regulating their expression or activities and not via changes in the ribosome.

It may even be possible that the causality here is reverse to what is proposed by the authors. It is possible that changes in gene expression cause changes in codon decoding rates because they alter the demand for tRNAs bearing amino acids enriched or depleted in these genes and subsequently change the availability of those tRNAs.

[Response] In order to address the reviewer's concern, we analyzed the expression level and aminoacylation status of tRNAs by Charged DM-tRNA-seq. In addition to sequencing tRNAs after demethylase treatment, this technique can also determine the aminoacylation status of tRNAs with a chemical step that specifically removes the 3' A residue in uncharged tRNAs. We detected a decrease in the abundance of tRNA-Pro-CGG and an increase in that of tRNA-Pro-TGG in the RPL3L-deficient heart, whereas the amounts of other tRNAs remained unchanged (**new Fig. 6a, b, and Supplementary Data 6**). Neither the rate of aminoacylation of tRNAs nor the protein level of aminoacyl-tRNA synthetases was found to be altered in the heart of RPL3L knockout mice relative to that of control mice (**new Fig. 6c, d, and Supplementary Data 6**). We next analyzed translation elongation dynamics for each codon in the RPL3L-deficient heart and found that translation elongation was delayed at all Pro and Ala codons (**new Fig. 3e, f**). These results suggest that the delay in elongation at the CCG codon is attributable in part to a decrease in tRNA-Pro-CGG expression, whereas that at other Pro and Ala codons is independent of the expression level or aminoacylation status of tRNAs.

As the reviewer points out, changes in tRNA modifications also affect translation elongation dynamics⁷, but the analysis of such modifications is technically challenging as a result of their complexity. On the other hand, given that m6A modification in the coding sequence is known to alter the secondary structure of certain mRNAs and to reduce ribosomal pausing⁸, we performed m6A-seq for the RPL3L-deficient heart. However, we found no change in the ratio of m6A for all transcripts (**new Fig. 6e, f, and Supplementary Data 7**) or in the abundance of proteins encoded by m6A-positive genes (**new Fig. 6g**) in the RPL3L-deficient heart compared with the control heart, suggesting that m6A status is not likely to contribute to altered translation dynamics induced by RPL3L loss.

As mentioned above, RPL3 is methylated at a histidine residue near the PTC by METTL18, and this methylation reaction slows the rate of elongation at Tyr codons. In addition, the W255C mutation in RPL3 markedly impairs peptidyl transferase activity. These findings suggest that differences in amino acids in the NH₂-terminal extension, basic thumb, and W finger regions between RPL3 and RPL3L may also affect peptidyl transferase activity and translation elongation dynamics. Further structural analysis will be required to characterize the actual structure of the RPL3L-ribosome and to determine how structural differences cause changes in translation elongation dynamics. We have now addressed these issues in the revised manuscript (page 10, lines 302-322).

[References]

7. Nedialkova DD, Leidel SA. Optimization of codon translation rates via tRNA modifications maintains proteome integrity. *Cell* **161**, 1606-1618 (2015).
8. Mao Y, *et al.* m(6)A in mRNA coding regions promotes translation via the RNA helicase-containing YTHDC2. *Nat. Commun.* **10**, 5332 (2019).

Shiraishi *et al.* Figure 6

Shiraishi *et al.* Figure 3

The moonlighting activity of RPL3L is also supported by the inhibition of its expression in the presence of exogenous RPL3L. Since it is RPL3L RNA and not just RPL3L protein that is being reduced, the inhibition ought to occur at RNA level suggesting that RPL3L can induce transcriptional changes when outside of the ribosome.

[Response] We apologize for a lack of clarity regarding this comment. RPL3L is not expressed in C2C12 cells, and we showed that forced expression of RPL3L in these cells resulted in a reduced abundance of *Rpl3* mRNA, not *Rpl3l* mRNA (original Fig. 4a). We now also show analysis of endogenous *Rpl3* mRNA in RPL3 OE cells with a primer set targeting the 3'UTR rather than the coding sequence. We found that the amount of endogenous *Rpl3* mRNA was also reduced by RPL3 overexpression (**new Fig. 4a**). Furthermore, a luciferase reporter assay revealed that *Rpl3* promoter activity was suppressed to the same extent in both RPL3 OE and RPL3L OE cells (**new Supplementary Fig. 5a**). We were unable to determine the promoter activity of *Rpl3l* because of the lack of RPL3L expression in C2C12 cells. These results thus indicate that RPL3L suppresses transcription of *Rpl3* outside the ribosome, but this suppressive activity is identical for RPL3 and RPL3L. We have now addressed this issue in the revised manuscript (page 8, lines 230-235).

Shiraishi et al. Figure 4

Shiraishi et al. Supplementary Figure 5

Technical questions/suggestions:
How ribosome occupancy was calculated?

[Response] We apologize for the lack of information on how to calculate ribosome occupancy in the Methods section. For the calculation of ribosome occupancy, we used 28- to 30-nt reads that showed 3-nt periodicity, and the A-site was defined as the position located 15 nt from the start of the reads. Ribosome occupancy was calculated as the average of the normalized read counts (RPM) corresponding to the positions of the E-, P-, and A-sites, and was further corrected by the average of the ribosome occupancy corresponding to all amino acids to obtain the relative ribosome occupancy. We have now clarified this point in the Methods section of the revised manuscript (page 22, lines 743-755).

Local decoding rates of translation on certain amino acids (Pro/Ala/Lys) is different between Rpl3l-/Rpl3l- with Rpl3l+/Rpl3l+ genotypes, but it makes sense to carry out this analysis for specific codons instead of on groups of synonymous codons. This will help to understand whether the effect is on amino acids, specific codons or tRNA isoacceptors .

Furthermore, the way how the local decoding rates (a.k.a. pauses, dwelling times) are

calculated could be very sensitive to the input of very few individual codons at which strong pauses occur at highly expressed mRNAs. This makes it difficult to detect global changes, since any changes in codon decoding rates of lowly expressed mRNAs would have negligible contribution to the signal. Thus, I strongly recommend to use RUST method (O'Connor et al DOI: 10.1038/ncomms12915). RUST greatly reduces the impact of individual locations and thus could help identification of global effects. I believe if the authors observation is genuine, RUST would provide a much stronger signal than what was obtained with the authors' method.

[Response] We analyzed ribosome occupancy at the codon level and found that it was increased at all Pro and Ala codons and decreased at all Lys and Asn codons in the RPL3L-deficient heart (**new Fig. 3e**). Further analysis with the use of RUST, as suggested by the reviewer, confirmed an increase in ribosome occupancy at all Pro and Ala codons, but failed to show a decrease at Lys and Asn codons (**new Fig. 3f**). RUST analysis also showed decreased ribosome occupancy at all Pro and Ala codons in C2C12 cells stably expressing RPL3L, but not in RPL3 OE cells (**new Supplementary Fig. 5c, d**). Given that the delayed translational elongation at Pro/Ala codons was a more consistent abnormality, we changed all data shown in Figure 5 of the original manuscript to focus only on Pro/Ala codons in the revised manuscript. We have now addressed these issues in the revised manuscript (page 7, lines 201-207, and page 8, lines 240-246).

Shiraishi et al. Figure 3

Shiraishi et al. Supplementary Figure 5

In Ribo-seq data processing, the authors excluded the first and the last five codons from the analysis. This is certainly a good idea, but perhaps a justification should be given for the readers unfamiliar with associated issues.

[Response] We excluded the first and last five codons from the analysis to avoid the atypical footprint counts observed around start and stop codons, as has been done by other groups previously^{9,10}. We have now clarified this point in the Methods section of the revised manuscript (page 22, lines 743-755).

[References]

- Iwasaki S, Floor SN, Ingolia NT. Rocaglates convert DEAD-box protein eIF4A into a sequence-selective translational repressor. *Nature* **534**, 558-561 (2016).
- Tunney R, McGlincy NJ, Graham ME, Naddaf N, Pachter L, Lareau LF. Accurate design of translational output by a neural network model of ribosome distribution. *Nat.*

Struct. Mol. Biol. **25**, 577-582 (2018).

Methods state that RNA fragments of size between 17 to 34 nt were kept although only 28 and 29nt reads were used with arbitrary 15nt offset. Could this be explained? What reads of other lengths were used for?

[Response] Given that we analyzed ribosome occupancy at each codon, only footprints that showed a 3-nt periodicity were used in our analysis. The 3-nt periodicity score was calculated by RibORF for each Ribo-seq sample, and footprints with a score of 0.5 or greater were applied to the analysis. Although Ribo-seq libraries were prepared with 17- to 34-nt footprints, given that footprints derived from ribosomes with open A-sites are 20 to 22 nt¹¹, 20- to 22-nt footprints were excluded from the analysis because most of them failed to show 3-nt periodicity in the libraries we generated in this study. Thus, only footprints of 28 to 30 nt that reproducibly showed 3-nt periodicity were used, and other footprints were excluded from our analysis. We have now clarified this point in the Methods section of the revised manuscript (page 22, lines 743-755).

[Reference]

11. Wu CCC, Zinshteyn B, Wehner KA, Green R. High-resolution ribosome profiling defines discrete ribosome elongation states and translational regulation during cellular stress. *Mol. Cell* **73**, 959-970 (2019).

The combined total number of disome peaks is given for both RPL3L-deficient and control hearts. But no information on their number and overlap between them is given for RPL3L-deficient and control samples.

[Response] As suggested by the reviewer, we have now provided data on the number and overlap of disome peaks for the RPL3L-deficient and control heart. Most peaks (652 peaks) were common to the RPL3L-deficient and control heart, with 25 and 45 peaks being detected specifically in the control heart and RPL3L-deficient heart, respectively (**new Fig. 7a**). We have now clarified this point in the revised manuscript (page 11, lines 333-335).

Shiraishi et al. Figure 7

Multiple testing correction needs to be applied to the p-values obtained for the comparison shown in Fig. 2i, and for correlation p-values on Fig 4e,f.

[Response] For Figure 4e and f, we have now performed the Bonferroni correction. However, for Figure 2i, we consulted a statistician, who told us that in this case a correction for multiple testing was not necessary. If we are mistaken, we would be happy for the

reviewer to advise us on the appropriate correction method.

Why the plot in Fig. 3b has two blue dots for RPL3L?

[Response] *Rpl3l* mRNA has two isoforms (isoform1, ENSMUST00000045186.10; isoform2, ENSMUST00000170239.8), with the two dots representing these two isoforms, as is now indicated in the **new Figure 3b** and its legend in the revised manuscript.

Shiraishi et al. Figure 3

On Fig. 5f colours for Pro and Ala are almost indistinguishable.

[Response] As mentioned above, we have removed the dots for Lys codons shown in Figure 5f of the original manuscript. The Pro and Ala codons are therefore now indicated in red and blue, respectively, in the revised figure.

Response to Reviewer #2

In this manuscript Shiraishi et al show that show that RPL3L, a paralog of RPL3, which is specifically expressed in heart and skeletal muscle, influences translation elongation dynamics. Deficiency of RPL3L-containing ribosomes, which the authors designated as “myo-ribosomes” in RPL3L knockout mice resulted in impaired cardiac contractility. The authors went on to examine the effects of deleting RPL3L in the mouse heart on mouse heart function under baseline conditions. They then went on to study the effects of RPL3L deletion in mouse hearts or overexpression in C2C12 myoblasts on translational elongation dynamics, ribosome occupancy and selective effects on translation of a subset of affected mRNAs.

Summary:

While this study represents an interesting area of ribosome function in translation in cardiac myocytes, there remain some methodological and other experimental issues that require attention and resolution, as described below. Moreover, there have been other publications, notably ref. 20, that do much of the same experiments; this decreases the novelty and impact of the current paper and, since different results were obtained in the ref. 20 from those described here, places Shiraishi et al in a position of describing why their results differ from those in ref. 20. Moreover, in ref 20 there were more conclusions drawn as to how RPL3L deletion might affect mitochondrial function in the heart, and in skeletal muscle as well.

[Response] We thank the reviewer for the careful evaluation of our manuscript. We also thank the reviewer for suggestions that we feel have helped us to greatly improve our manuscript. Our specific responses to the points raised are as follows (this response contains low-resolution thumbnails for clarity; please refer to the manuscript for high-resolution figures):

Comments:

1- I don't think the authors should use the term “myo-ribosomes”, but instead should use the more accurate term, RPL3L-ribosomes, because there is a possibility that other myocyte specific ribosomal proteins will be discovered and that the ribosomes they are part of might not be the same ribosomes as the RPL3L-containing ribosomes.

[Response] As suggested by the reviewer, we have changed all mentions of “myo-ribosomes” to “RPL3L-ribosomes” in the revised manuscript.

2- One of the limitations of the study that the authors need to address is that the ribo-tag and RNA-seq are examining mRNAs from all cell types in the heart, while the deletion affects only a subset of the cell types, i.e. some myocytes.

[Response] We applied a primary culture system to enrich cardiomyocytes (CMs), but the number of CMs obtained by this method is limited and the cells undergo cell death during culture¹², which may be preceded by impairment of their translational activity. A high 3-nt periodicity score and abundant reads are required for analysis of ribosome occupancy at each codon, but the Ribo-seq data obtained from the primary cultured CMs failed to satisfy these requirements.

The amount of ribosomes would be expected to be correlated with cell volume. The mammalian heart is composed of many cell types, with the most common being CMs, fibroblasts (FBs), endothelial cells (ECs), and perivascular cells. In the human and rat heart, CMs constitute 30% to 40% of total cells, but the average volume of CMs is 20 to 25 times that of FBs and ECs, with the result that CMs occupy 70% to 85% of the tissue by volume¹³. Given that the number of CMs is even greater in the mouse heart (54% of total cells) than in the human and rat heart¹⁴, most of the Ribo-seq footprints obtained from the mouse heart are likely to be derived from the ribosomes of CMs.

[References]

12. Ackers-Johnson M, Li PY, Holmes AP, O'Brien SM, Pavlovic D, Foo RS. A simplified, langendorff-free method for concomitant isolation of viable cardiac myocytes and nonmyocytes from the adult mouse heart. *Circ. Res.* **119**, 909-920 (2016).
13. Zhou PZ, Pu WT. Recounting cardiac cellular composition. *Circ. Res.* **118**, 368-370 (2016).
14. Banerjee I, Fuseler JW, Price RL, Borg TK, Baudino TA. Determination of cell types and numbers during cardiac development in the neonatal and adult rat and mouse. *Am. J. Heart Circ. Physiol.* **293**, H1883-H1891 (2007).

3- While the structural analysis shown in 1f and 1g are interesting, they really aren't critical to the study and may suggest more than the authors want to infer. For example, based on the overlay of RPL3 and RPL3L, the reader is led to believe that these proteins occupy the same site in the ribosome and I don't think the authors know enough to know whether that is true.

[Response] As the reviewer points out, the structure of RPL3L is just a prediction by AlphaFold2 and might be misleading. We have therefore modified the text in the revised manuscript to emphasize this point, and we now state that further detailed biochemical and structural analysis is warranted to characterize the actual structure of the RPL3L-ribosome and to determine how structural differences might cause changes in translation elongation dynamics (page 5, lines 130-131, and page 14, lines 447-449).

Given that the structural prediction was performed by our authors who are experts in cryo-EM analysis of ribosomes¹⁵, and Reviewer #3, who is an expert in structural analysis, did not raise any concerns about our structure data, we believe that our structural prediction has been performed properly.

We believe that structural prediction for RPL3L is important to infer its function. RPL39L, which is specifically expressed in sperm, was recently shown to contribute to cell type-specific regulation of translation elongation⁶. Cryo-EM analysis of RPL39L-containing ribosomes revealed that RPL39L is located in the exit tunnel of nascent proteins, at the same position as RPL39. The exit tunnel of RPL39L-containing ribosomes differs from that of RPL39-containing ribosomes in both size and charge state. RPL39L-containing ribosomes regulate cotranslational folding of a subset of male germ cell-specific proteins essential for spermatogenesis, and ablation of RPL39L results in increased protein aggregation associated with abnormal folding, rendering male mice infertile. Although it remains unclear whether RPL3L is located at the same position as RPL3, the structure prediction suggests that differences in amino acids in the NH₂-terminal extension, basic thumb, and W finger regions between RPL3 and RPL3L may affect peptidyl transferase activity and translation elongation dynamics.

[References]

15. Yokoyama T, *et al.* HCV IRES captures an actively translating 80S ribosome. *Mol. Cell* **74**, 1205-1214 (2019).
6. Li H, *et al.* A male germ-cell-specific ribosome controls male fertility. *Nature* **612**, 725-731 (2022).

4- The effect of RPL3L deletion on mouse heart function is subtle and mild. It would be important to know whether the functional effect is exacerbated under a typical cardiac stress situation, such as pressure overload. In this regard, a table of the complete echocardiography parameters should be included.

[Response] As suggested by the reviewer, we subjected the RPL3L knockout mice to pressure overload. Although transverse aortic constriction (TAC) burden and angiotensin II (Ang II) stimulation are generally applied to induce cardiac pressure overload, we used Ang II infusion with a micro-osmotic pump, which is less invasive and less susceptible to technical errors. Cardiac hypertrophy, especially of the left ventricle, was observed after Ang II infusion, confirming the successful induction of pressure overload, but there was no significant difference in the change in heart weight induced by Ang II treatment between RPL3L-deficient and control mice (**new Supplementary Fig. 3a**). In addition, whereas LVEF and LVFS were reduced in RPL3L-deficient mice compared with control mice in the saline infusion group, these significant differences were no longer observed in the Ang II infusion group (**new Supplementary Fig. 3b, c**). These results indicate that the impairment of cardiac contractility in mice lacking RPL3L was not exacerbated by pressure overloading with Ang II stimulation. Although mutations in the *RPL3L* gene have been identified in patients with atrial fibrillation or pediatric dilated cardiomyopathy, RPL3L-deficient mice did not reveal such pronounced abnormalities even under the pressure overload condition, possibly because of other species differences or interactions with other genes. We have now addressed this issue in the revised manuscript (page 6, lines 173-183), with all echocardiography parameters now being listed in **new Supplementary Data 1**.

Shiraishi et al. Supplementary Figure 3

5- While it looks convincing that RPL3L deletion changes translational dynamics, it is unclear whether any of these changes contributes to the contractility phenotype seen in the mice, in vivo.

6- Related to the comment above, it would be important for the authors to select specific proteins whose expression was affected by RPL3L deletion and to list them and discuss the possible implications. The GO categories are a start, but drilling down deeper is important for readers to interpret the data.

[Response] As suggested by the reviewer, we have now added a more in-depth discussion of the results of GO analysis. The group of genes with the GO term “cardiac muscle contraction” ($n = 46$) was further explored along with the categories of “dilated cardiomyopathy (DCM)-related genes”¹⁶ ($n = 33$), “genes with differential ribosome

occupancy at Pro/Ala codons in the heart of *Rpl3l*^{+/+} or *Rpl3l*^{-/-} mice (Pro/Ala)” (*n* = 44), and “genes with disome peaks in the heart of *Rpl3l*^{+/+} or *Rpl3l*^{-/-} mice (disome-positive)” (*n* = 198). A Venn diagram showing the overlap of these four groups of genes is now included as **new Figure 8a**. Approximately one-third of genes in the “cardiac muscle contraction” group are included in the “disome-positive” group (*n* = 14), with these overlapping genes also including genes in the “Pro/Ala” group (*n* = 5). These 14 genes include those for actin (*ACTC1*, encoding cardiac α -actin) and myosin (*MYH6*, *MYH7*, *MYL2*, *MYL3*, and *MYL4*, encoding cardiac myosins) fibers that align to form sarcomeres, those for tropomyosin (*TPMI*, encoding the tropomyosin α 1 chain) and troponin (*TNNC1*, *TNNI3*, and *TNNT2*, encoding each of the three troponin subunits), *MYBPC3* (encoding a protein that binds to actin and myosin and thereby enhances cardiac contractility)¹⁷, *TCAP* (encoding a protein that cross-links Titin, a filamentous protein with springlike properties in the I-band region, to the Z-disc)¹⁸, *CSRP3* (encoding a scaffold protein involved in multiple protein-protein interactions within the Z-disc)¹⁹, and *CASQ2* (encoding a calcium-binding protein that localizes to the sarcoplasmic reticulum to store calcium)²⁰. In addition, the overlapping genes between the “DCM-related” and “disome-positive” groups also include *LDB3*, which encodes a PDZ-LIM domain-binding factor that provides structural stability to the Z-disc²¹. The cumulative fraction for the fold change in protein expression in the heart of *Rpl3l*^{-/-} mice compared with that of control mice revealed that the abundance of proteins encoded by these 15 genes (enclosed by the red boundaries in Figure 8a, but only 12 proteins were detected by MS) was significantly reduced in the heart of *Rpl3l*^{-/-} mice relative to all proteins (*n* = 3376) ($p = 3.03 \times 10^{-5}$) (**new Fig. 8b**). These results suggest that downregulation of these gene products is likely responsible for the cardiac contraction defect in *Rpl3l*^{-/-} mice.

We applied a similar analysis to nuclear genes for oxidative phosphorylation complexes in the inner mitochondrial membrane that are responsible for ATP production (*n* = 91). The protein subunits of these complexes are encoded in both the nuclear and mitochondrial genomes, with their transcripts being translated by cytoplasmic ribosomes (including RPL3L-ribosomes) and mitochondrial ribosomes, respectively. Many of these proteins encoded in the nuclear genome were also found to be associated with delayed translation elongation at Pro/Ala codons or disome peaks (**new Fig. 8c**), but these proteins failed to show a significant decrease in expression in the RPL3L-deficient heart, showing only such a tendency (**new Fig. 8d**).

Collectively, these results suggest that reduced levels of proteins associated with cardiac muscle contraction, rather than of those associated with the mitochondrial respiratory chain, are likely responsible for the impaired cardiac contractility observed in RPL3L-deficient mice. We have now addressed these issues in the revised manuscript (pages 11-12, lines 349-391).

[References]

16. Dellefave L, McNally EM. The genetics of dilated cardiomyopathy. *Curr. Opin. Cardiol.* **25**, 198-204 (2010).
17. Previs MJ, *et al.* Myosin-binding protein C corrects an intrinsic inhomogeneity in cardiac excitation-contraction coupling. *Sci. Adv.* **1**, e1400205 (2015).
18. Zou PJ, *et al.* Palindromic assembly of the giant muscle protein titin in the sarcomeric Z-disk. *Nature* **439**, 229-233 (2006).
19. Chauhan PK, Sowdhamini R. LIM domain-wide comprehensive virtual mutagenesis provides structural rationale for cardiomyopathy mutations in CSRP3. *Sci. Rep.* **12**, 3562 (2022).
20. Rossi D, Gamberucci A, Pierantozzi E, Amato C, Migliore L, Sorrentino V.

Calsequestrin, a key protein in striated muscle health and disease. *J. Muscle Res. Cell Motil.* **42**, 267-279 (2021).

21. von Nandelstadh P, *et al.* A class III PDZ binding motif in the myotilin and FATZ families binds enigma family proteins: a common link for Z-disc myopathies. *Mol. Cell Biol.* **29**, 822-834 (2009).

Shiraishi *et al.* Figure 8

7- C2C12 myoblasts are not a good model system with which to examine RPL3L overexpression (OE), as the authors have done. In fact, in the Introduction the authors state that “Forced expression of RPL3L in C2C12 myogenic cells inhibits myoblast fusion and myotube growth¹⁹, whereas loss of RPL3L increases ribosome-mitochondria interactions in mouse cardiomyocytes²⁰.” The inhibition of myoblast fusion and myotube growth would seem to be antithetical to the activity the authors would like to ascribe to RPL3L.

[Response] The experiments mentioned by the reviewer (forced expression of RPL3L in C2C12 cells²²) were conducted under the differentiation condition. In contrast, our Ribo-seq analysis of C2C12 myoblasts was performed under the proliferative condition. To examine further the effects of RPL3 or RPL3L overexpression under the proliferative condition, we performed GO analysis of genes upregulated in RPL3L OE cells compared with RPL3 OE cells using RNA-seq data. We found that genes for “regulation of muscle contraction” and “skeletal muscle contraction” were upregulated in RPL3L OE cells compared with RPL3 OE cells (**new Supplementary Fig. 6a**), consistent with our results obtained with the RPL3L-deficient heart.

Of note, we found that genes for “skeletal muscle cell differentiation” were also upregulated in RPL3L OE cells compared with RPL3 OE cells (**new Supplementary Fig. 6a**), which appears inconsistent with the findings of the previous study showing that overexpression of RPL3L in C2C12 myoblasts inhibits myoblast fusion and myotube

growth²². Indeed, we observed significant inhibition of both myoblast fusion and myotube growth in RPL3 OE cells compared with control cells, but such inhibition was not apparent in RPL3L OE cells (**new Supplementary Fig. 6b, c**). These results indicate that overexpression of RPL3, but not of RPL3L, suppresses myoblast differentiation. These lines of evidence obtained with C2C12 myoblasts, including all data in Figure 4 of the original manuscript, further corroborate our findings obtained with mice lacking RPL3L. We have now addressed these issues in the revised manuscript (pages 8-9, lines 258-275).

[Reference]

22. Chaillou T, Zhang XP, McCarthy JJ. Expression of muscle-specific ribosomal protein L3-like impairs myotube growth. *J. Cell. Physiol.* **231**, 1894-1902 (2016).

Shiraishi et al. Supplementary Figure 6

8- In ref. 20 in the current manuscript there is a more complete description of how RPL3L deletion “modulates the expression of RPL3, which in turn affects ribosomal subcellular localization and, ultimately, mitochondrial activity”, which better describes the functional consequences of RPL3L depletion than does the current paper. In fact, the current paper’s finding that cardiac function/contractility was affected is not what was seen in the paper described in ref. 20, where the authors stated that “echocardiographic profiles of Rpl3l-/- mice showed no significant differences to those of control Rpl3l+/+.”

[Response] Milenkovic and coauthors posted an analysis of RPL3L-deficient mice in the preprint server bioRxiv, but they did not find impaired cardiac contractility in young (8-week-old) mutant mice²³. In contrast, we analyzed 18- to 21-week-old mice, as well as 20- to 27-week-old mice in the case of the new pressure overload experiments, by echocardiography. Given that heart failure generally progresses chronically, we speculate

that Milenkovic *et al.* may have failed to detect late-onset cardiac phenotypes. In addition, Grimes and coauthors have recently generated RPL3L-deficient mice and found that the heart of 18-month-old mutant mice is significantly smaller than that of controls without apparent defects in cardiac contraction²⁴. The C57BL/6NTac substrain was used in the analysis by Grimes *et al.*, whereas the C57BL/6J substrain was applied in the analysis by Milenkovic *et al.* as well as by us, suggesting that this genotypic difference may be responsible for the differences in cardiac phenotypes. Milenkovic *et al.* showed that loss of RPL3L increased ribosome-mitochondria interactions with elevated mitochondrial activity, but Grimes *et al.* failed to detect any defects in ribosome localization, ultrastructure, or mitochondrial function in the cardiac muscle of RPL3L-deficient mice.

Milenkovic and coauthors also performed Ribo-seq analysis for the heart of RPL3L-deficient mice, but they did not find a difference in ribosome occupancy at the A-site. We do not know the exact reason for this, but in general ribosomes are highly sensitive to stress during isolation and it is relatively challenging to obtain high-quality data that are required for ribosome occupancy analysis at each codon, especially with tissue as a ribosome source. Our Ribo-seq data showed high 3-nt periodicity (Supplementary Fig. 4a, b, in the revised manuscript), and delayed translation elongation at Pro/Ala codons was reproducibly observed with both our algorithm and the newly applied RUST algorithm (**new Fig 3f**). In addition, translation dynamics in C2C12 cells stably expressing RPL3L were highly inversely correlated with those in the RPL3L-deficient heart, whereas no change at Pro/Ala codons was observed in C2C12 cells stably expressing RPL3. Moreover, decreased protein levels were observed in the RPL3L-deficient heart for genes with delayed elongation at Pro/Ala codons and for genes prone to ribosome collisions. This series of consistent results demonstrates the high reliability of our findings. We have now clarified these points in the Discussion section of the revised manuscript (pages 14-15, lines 450-480).

[Reference]

23. Milenkovic I, *et al.* Dynamic interplay between RPL3- and RPL3L-containing ribosomes modulates mitochondrial activity in the mammalian heart. *bioRxiv*, 2021.2012.2004.471171 (2023).
24. Grimes KM, *et al.* Rpl3l gene deletion in mice reduces heart weight over time. *Front. Physiol.* **14**, 1054169 (2023).

Shiraishi *et al.* Supplementary Figure 4

Shiraishi *et al.* Figure 3

Response to Reviewer #3

In the present paper, Shiraishi and colleagues investigate why in heart and skeletal muscle RPL3L, the paralog of canonical uL3, is overexpressed. Overall, tissue-specific expression of paralogs of ribosomal proteins and how this regulates and fine-tunes gene expression is an enigmatic and important phenomenon.

Here, the authors show that in the heart of wt mice about two thirds of ribosomes are so-called myo-ribosomes and have RPL3L instead of uL3. RPL3L-deficient knock-out mice overexpress uL3 to compensate but suffer from impaired cardiac contractility establishing that the loss of myo-ribosomes has specific physiological consequences. Further analysis using Ribo-seq shows that the lack of myo-ribosomes affects translation elongation dynamics for certain codons (delayed at Pro and Ala codons, increased at Asn and Lys codons) and results in increased ribosome collisions/ disome formation. Altered translation elongation dynamics in turn results in reduced expression of proteins related to mitochondrial ATP synthesis and cardiac muscle contraction.

Accordingly, Shiraishi and colleagues demonstrate that RPL3L-containing myo-ribosomes are important for the efficient expression of specific proteins. Whereas the exact molecular mechanism of RPL3L remains unclear, the paper rationalizes the physiological importance of the ribosomal protein paralog and paves the way for further in-depth biophysical and structural analysis. Overall, the paper is well written. I have only some suggestions to improve the paper.

[Response] We thank the reviewer for the positive comments on our manuscript. The reviewer states that “*the paper rationalizes the physiological importance of the ribosomal protein paralog and paves the way for further in-depth biophysical and structural analysis*” and that “*Overall, the paper is well written.*” We also thank the reviewer for suggestions, which we feel have helped us to improve our manuscript. Our specific responses to the points raised are as follows (this response contains low-resolution thumbnails for clarity; please refer to the manuscript for high-resolution figures):

- 1. It may be helpful to highlight the position of amino acids that differ between uL3 and RPL3L (Fig. 1h) also in the structural model depicted in Fig. 1g.*

[Response] As suggested by the reviewer, the amino acids that differ between RPL3 and RPL3L have been highlighted in orange with their amino acid numbers (**new Fig. 1g**). This makes it clearer to the readers which amino acid positions in the three-dimensional structure differ between RPL3 and RPL3L.

Shiraishi et al. Figure 1

2. The authors write that no significant differences for other RPs were detected (page 6, 1st sentence) and refer to extended Data Fig 2c. What about RPS15?

[Response] Although RPS15 abundance tended to be increased in the RPL3L-deficient heart relative to the control heart, this difference was not significant. It is also unlikely that the loss of RPL3L would affect the amount of RPS15, given the distant separation of these two proteins in the three-dimensional structure. We have now clarified this point in the revised manuscript (page 6, lines 155-158).

3. In the discussion, the authors speculate that increased flexibility of the PTC region may led to a reduced efficiency of peptidyl transfer. However, the loss of myo-ribosomes differentially affects elongation efficiency dependent on the amino acid incorporated. Thus, the very entrance the tunnel for the nascent peptide could be affected as well.

[Response] As suggested by the reviewer, RPL3 and RPL3L may affect not only the efficiency of peptide translocation, but also the entrance of the nascent peptide tunnel. We have now added this point to the Discussion section of the revised manuscript (page 14, lines 444-447).

REVIEWERS' COMMENTS

Reviewer #1 (Remarks to the Author):

The authors have addressed all my comments thoroughly and responsibly. I am fully satisfied with their response and have only minor suggestions:

1. Regarding multiple test correction, I am not a statistician myself, however, it seems to me that the multiple testing correction is warranted for 4e, f. The more different characteristics we compare the more likely one of them would turn out to be statistically significant (pass the default significance threshold) by chance, the point effectively made by the green jelly cartoon:

<https://sites.uab.edu/periop-datascience/2020/07/29/more-p-values-mode-problems/>

Thus, I would recommend doing multiple testing correction.

2. When discussing the effects of RPL3 and RPL3L on their mRNA levels the authors describe it as "transcription suppression". Mechanistically this could also be via mRNA stability, so it may be better to describe it as "mRNA levels suppression" or something similar.

Pasha Baranov

Reviewer #2 (Remarks to the Author):

In this revised manuscript the authors have addressed most of my previous comments sufficiently. In particular, they addressed the question about the RPL3L mouse phenotype with overlaid on a stress, in their case treatment with AngII, which represents a significant amount of work on the authors' part. The data look very convincing, as the amount of hypertrophy achieved is what is expected, even if RPL3L deletion did not change the response, it was something that was important to have determined. I also appreciate that the authors took a deeper dive into dissecting and analyzing what genes that are altered in the RPL3L deletion mouse could account for the decreased contractility. Also, the authors addressed a recent publication where the same gene was deleted in mouse hearts, and a somewhat different phenotype was obtained. Again, this was important to address, as the authors did now in the Discussion, so that readers can evaluate why the two papers came up with different results. The authors also revised the title of the manuscript in accord with my comment on that, which is much appreciated. In summary, I appreciate the efforts the authors have gone to, to address my comments and commend them on a revision job well done.

Reviewer #3 (Remarks to the Author):

The authors have thoroughly addressed the reviewers' comments. I do not have further comments.

Response to Referee 1

We thank Dr. Baranov again for the constructive suggestions that we feel have helped us to greatly improve our manuscript. Our specific responses to the points raised are as follows:

1. Regarding multiple test correction, I am not a statistician myself, however, it seems to me that the multiple testing correction is warranted for 4e, f. The more different characteristics we compare the more likely one of them would turn out to be statistically significant (pass the default significance threshold) by chance, the point effectively made by the green jelly cartoon:

<https://sites.uab.edu/periop-datascience/2020/07/29/more-p-values-mode-problems/>

Thus, I would recommend doing multiple testing correction.

[Response] We agree with the reviewer that multiple test corrections are necessary for Fig. 4e and f. We have performed the Bonferroni corrections for the data in the previous revised manuscript.

2. When discussing the effects of RPL3 and RPL3L on their mRNA levels the authors describe it as “transcription suppression”. Mechanistically this could also be via mRNA stability, so it may be better to describe it as “mRNA levels suppression” or something similar.

[Response] As suggested by the reviewer, we stated in the revised manuscript that “RPL3L suppresses the expression level of *Rpl3* mRNA” (page 8, lines 234-235).

Response to Referee 2

We thank the reviewer for his/her positive comments. During this review, the manuscript posted on the preprint server bioRxiv was published in Nucleic Acids Research and its reference information has been updated in the revised manuscript (page 25, lines 856-858). This reviewer has no more suggestions.

Response to Referee 3

We thank the reviewer again, and this reviewer has no more suggestions.